

# The holographic nature of null infinity

**Alok Laddha**[1][⋆], **Siddharth G. Prabhu**[2][†], **Suvrat Raju**[2][‡] **and Pushkal Shrivastava**[2][∘]

**1** Chennai Mathematical Institute, Siruseri, Chennai, India.
**2** International Centre for Theoretical Sciences, Tata Institute of Fundamental Research,
Shivakote, Bengaluru 560089, India.

⋆ aladdha@cmi.ac.in, † siddharth.prabhu@icts.res.in,
‡ suvrat@icts.res.in, ∘ pushkal.shrivastava@icts.res.in

## Abstract

We argue that, in a theory of quantum gravity in a four dimensional asymptotically flat spacetime, all information about massless excitations can be obtained from an infinitesimal neighbourhood of the past boundary of future null infinity and does not require observations over all of future null infinity. Moreover, all information about the state that can be obtained through observations near a cut of future null infinity can also be obtained from observations near any earlier cut although the converse is not true. We provide independent arguments for these two assertions. Similar statements hold for past null infinity. These statements have immediate implications for the information paradox since they suggest that the fine-grained von Neumann entropy of the state defined on a segment $(-\infty, u)$ of future null infinity is independent of $u$. This is very different from the oft-discussed Page curve that this entropy is sometimes expected to obey. We contrast our results with recent discussions of the Page curve in the context of black hole evaporation, and also discuss the relation of our results to other proposals for holography in flat space.

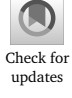

# 1 Introduction

The principle of holography has been understood for asymptotically AdS spacetimes for more than twenty years, but some of its implications for how quantum information is stored in theories of quantum gravity are still not widely appreciated.

For asymptotically AdS spaces, holography states that a bulk theory of gravity is described by a dual conformal field theory [1–3]. But, crucially, the "extrapolate dictionary" [4] suggests that operators in the boundary theory are just given by taking the asymptotic limit of bulk operators. If we take this dictionary seriously, it leads to a remarkable prediction that can be stated purely in the bulk theory of gravity, without any reference to the CFT: since boundary operators, at a single instant of time, have complete information about the quantum state, asymptotic operators in the bulk theory on *any time slice* also have all information about the bulk state.

So, consider a Cauchy slice through the bulk, which may contain black holes or other excitations in its interior. The claim is that the degrees of freedom on the boundary of the same slice already carry complete information about these excitations. This is so surprising, and so unlike how quantum information is stored in a usual quantum field theory, that even practitioners of holography tend to often forget this fact, or at least wish it away by classifying it as something that should be ignored for all practical purposes!

However, it was recently argued in [5], following earlier work [6, 7], that if one thinks carefully about canonical gravity with asymptotically AdS boundary conditions, it is indeed possible to conclude that any two wavefunctions that are distinct in the bulk can also be distinguished by asymptotic operators.

The object of this paper is to understand how quantum information is stored holographically in flat space. The boundary of flat space is null infinity, and we have found that it stores information in a more intricate manner than the timelike boundary of AdS. We will argue that the following results hold for quantum gravity in four dimensional flat space.

1. All information about massless excitations in a quantum state — which one might naively have thought requires observations over all of future null infinity ($\mathcal{I}^+$) — can be obtained from an infinitesimal neighbourhood of null infinity near its past boundary ($\mathcal{I}^+_-$).

2. On future null infinity, any information about the quantum state that is available in the neighbourhood of a cut is also available in the neighbourhood of any cut to its past.

Precisely analogous statements hold for past null infinity, $\mathcal{I}^-$. The analogue of result (1) is that all information about massless excitations is available in an infinitesimal interval near its future boundary ($\mathcal{I}^-_+$). The analogue of result (2) is that information available in the neighbourhood of a cut of $\mathcal{I}^-$ is also available in any cut to its future.

These results should be understood in the following sense. Currently, we do not know how to define quantum gravity nonperturbatively about flat space. The claim is that any UV completion of the semiclassical theory should obey the results above, subject to certain reasonable physical assumptions that we now outline.

The assumptions that go into result (1) are that, in the full UV-complete theory of quantum gravity, (a) the vacua can be identified by operators near the past boundary of $\mathcal{I}^+$ (b) operators near the past boundary of $\mathcal{I}^+$ can map any vacuum to any other and (c) the spectrum of the Hamiltonian in the full theory remains bounded below. Assumption (a) is clearly true in semiclassical gravity, where the vacua are ground states of the ADM Hamiltonian, labeled by supertranslation charges, all of which can be defined near the past boundary of $\mathcal{I}^+$. Assumption (b) can also be checked explicitly. Since assumptions (a) and (b) are both about the vacua of the theory, they simply state that the UV-complete theory shares some of the low-energy structure of the semiclassical theory. Assumption (c) is difficult to prove but we do expect it to hold in any reasonable UV-complete theory of gravity.

The reader will note that result (2) above is stronger than result (1). In fact, result (1) follows as a special limit of result (2). This is because the assumptions that go into result (2) are also stronger: this result requires us to assume that certain commutation relations that can be derived at null infinity in the semiclassical theory are corrected only by local terms in the full theory of quantum gravity. Although we argue that this assumption is true at all orders in perturbation theory, we do not know how to justify it nonperturbatively. However, we note that even if result (2) fails nonperturbatively, this does not affect the validity of result (1).

The results above have immediate implications for the black hole information paradox. A tremendous amount of attention has been focused on the question of how the information "emerges" from the black hole as it evaporates. However, our results suggests that this is *not* the right question to ask. Rather, our results suggest that the information is *always* available outside. This can be formalized in terms of the next two important results of our paper

3. The von Neumann entropy of any pure or mixed state of massless excitations, defined on a segment $(-\infty, u)$ of future null infinity, is independent of the upper limit $u$.

4. The von Neumann entropy of any state defined on a segment $(u_1, u_2)$ of future null infinity, with $u_2 > u_1$, is independent of $u_2$.

Analogous results hold for $\mathcal{I}^-$. The analogue of result 3 is that the von Neumann entropy of any state of massless excitations on a segment $(v, \infty)$ is independent of the lower limit $v$ on $\mathcal{I}^-$; the analogue of result 4 is that the von Neumann entropy of a segment $(v_1, v_2)$ is independent of its lower limit.

We show that result (3) follows directly from result (1). Result (4), which is stronger than result (3), follows directly from result (2) and therefore also relies on the stronger assumptions that enter result (2). The main results of our paper, together with the assumptions that they are based on, are displayed in Fig 1.

These results imply that the oft-discussed Page curve for the von Neumann entropy of the black hole radiation at future null infinity is *not* the right expectation. Rather, since all the information about massless particles is already available near the extreme past of future null infinity, no new information becomes available as we move along null infinity towards the future. So the von Neumann entropy of the state defined on null infinity remains constant![1] It

---

[1]As we describe in section 4, this constant may not be 0 even in a unitary theory because our analysis does not currently include massive particles.

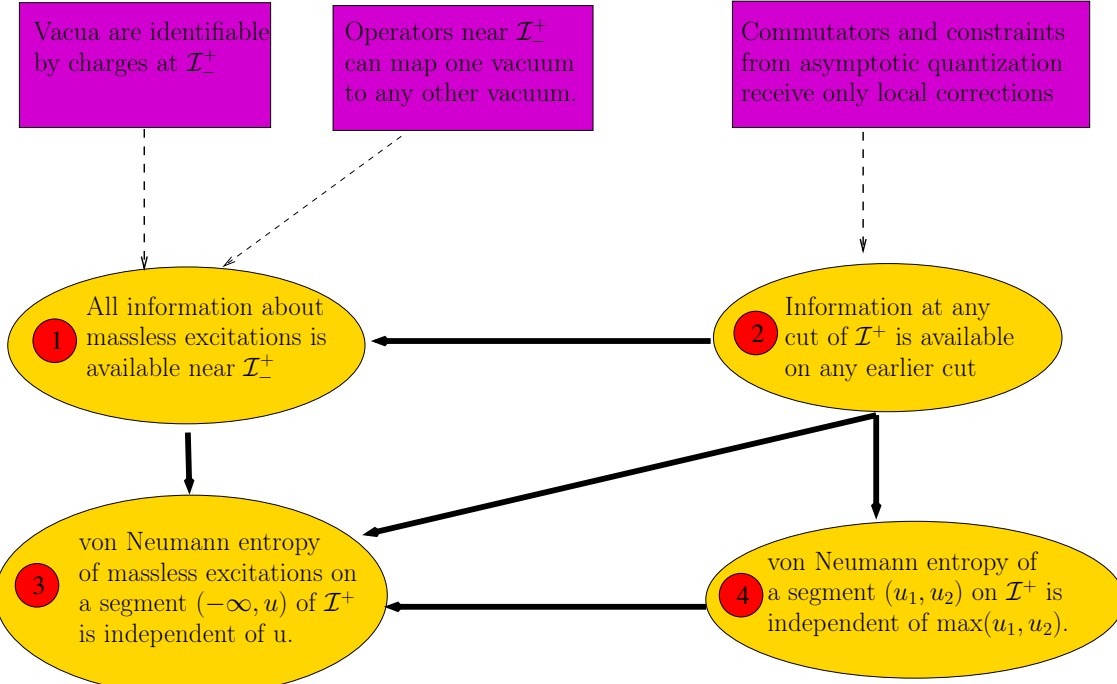

Figure 1: The main results of our paper are displayed in yellow ovals, with numbers in adjoining circles. The purple rectangles indicate physical assumptions. Dependence of a result on an assumption is indicated by a dashed line. Implications between results are denoted by thick black lines.

is not surprising that the Page curve does not apply to black hole radiation since the Page curve is derived by considering systems where the Hilbert space factorizes neatly into two parts. This assumption has been known to be wrong in quantum gravity.

Our results are not in contradiction with the recent derivations of the Page curve in AdS/CFT [8–11]. The reason is that the setups considered in those papers involve a factorized Hilbert space, which is achieved by turning gravity off at some point, and then considering the entanglement between the gravitational and the non-gravitational system. But our results do suggest that those discussions of the Page curve are not immediately relevant to the calculation of the evolution of von Neumann entropy of the black hole exterior in our world, where gravity does not get turned off sharply at any point.

There has been considerable work on understanding holography in asymptotically flat spacetimes starting with the work of de Boer and Solodukhin [12–104]. But this paper is complementary to much of this literature since our focus is neither on rewriting S-matrix elements as correlators in a dual theory, nor on the question of asymptotic symmetries. Rather we are interested in the novel question of how quantum information is *stored* at null infinity. Our work is closest in spirit to some of the earlier work of Marolf [105–107] but both our arguments and results are somewhat different as will be evident below.

This paper is organized as follows. In section 2, we describe our setup. Section 3 contains the main results of this paper—where we state and establish our first two results, subject to the assumptions explained above. Section 4 describes the relevance of these results for the information paradox. We discuss some subtleties and future open questions in section 5. Since the treatment of past and future null infinity is symmetric in our analysis, we will often make reference only to future null infinity, but this is only to avoid duplicating all statements.

Appendix A reviews the argument for the holographic storage of quantum information in spacetimes with asymptotically AdS boundary conditions. While this reviews material that

may be found in [5], we sharpen some of the assumptions of [5] and also clarify some of the reasoning. If the reader is unfamiliar with these arguments, we urge her to read this self-contained Appendix, since it may help to explain the logic of the main text of the paper in a more-familiar and simpler setting.

In Appendix B, we address the argument made in [108] that asymptotic charges should not be observables at null infinity in the quantum theory.

## 2 Background

In this section we review the basic ideas that underlie the physics of 4-dimensional asymptotically flat spacetimes. Our emphasis will be on those concepts that are most relevant to our analysis and for a more detailed analysis, the reader can consult a number of extensive reviews including [109–111]. We will first review the relevant boundary conditions and constraints on the dynamical fields at null infinity. Ashtekar [112] first noticed the remarkable fact that if one considers the covariant phase space of gravity as parameterized by data on null infinity, then even in the *full nonlinear theory*, one obtains a remarkably simple symplectic structure. Upon quantization, this structure leads to a Hilbert space that is the direct sum of an infinite number of "soft" sectors, and this has received renewed attention in light of the conservation laws associated to BMS symmetry [113–116].

### 2.1 Boundary conditions and constraints at null infinity

The space of all asymptotically flat space-times can be parameterized in terms of the retarded Bondi co-ordinates ($u = t - r,\ r,\ \Omega = (\theta, \phi)$) as

$$ds^2 = -du^2 - 2du\,dr + r^2\gamma_{AB}d\Omega^A d\Omega^B + rC_{AB}d\Omega^A d\Omega^B + \frac{2m_{\mathrm{B}}}{r}du^2 + \gamma^{DA}D_D C_{AB}du\,d\Omega^B + \dots,\quad (1)$$

where $\gamma_{AB}$ is the unit metric on $S^2$, with the corresponding derivative operator $D_A$. $C_{AB}(u,\ \Omega)$ is known as the shear field and contains complete information about the radiative degrees of freedom. In the Bondi frame, $C_{AB}$ is trace free, $\gamma^{AB}C_{AB} = 0$. This radiative data, in fact, "lives at" future null infinity $\mathcal{I}^+ := S^2 \times \mathbf{R}$ with coordinates $(\Omega, u)$. Viewed in this way, $S^2$ is a sphere at null infinity with the intrinsic metric $\gamma_{AB}$, and is known as the celestial sphere. We will denote the $S^2$ that lives at $u \to -\infty$ by $\mathcal{I}^+_-$. $m_{\mathrm{B}}(u,\ \Omega^A)$ is called the Bondi mass aspect.

In classical discussions of asymptotically flat spacetimes, a distinction is often made between spacetimes that contain black holes and those that do not. However, since we will be interested in the quantum theory, where all black holes are expected to eventually evaporate, we will make no such distinction. In particular, the future conformal boundary of our spacetime will always be $\mathcal{I}^+$.

In the presence of massless fields, fall-off conditions on these fields are dictated by the fact that total energy leaking out at null infinity is finite. For example, for a free massless scalar field, the large-$r$ expansion in Bondi co-ordinates is given by,

$$\phi(r, u, \Omega) = \frac{1}{r}O(u,\ \Omega) + O(\frac{1}{r^2}).\quad (2)$$

We will refer to $O$ (and the corresponding leading $\frac{1}{r}$ coefficients for other fields) as radiative matter data. When the field is coupled to gravity, this expansion is no longer valid and the leading classical term falls off as $\frac{\ln r}{r}$ [117–119]. This modification arises as the free radiative data $O$ is dressed by metric perturbations. (In the classical theory, these metric components are themselves sourced by $O$.) However even in this context, the independent data is still given by $O$ and the radiative degrees of freedom of the metric.

The components of the metric and the matter fields at null infinity are not all independent. The rate of change of the Bondi mass aspect $m_B(u, \Omega)$ is determined in terms of the radiative data via the following constraint

$$\partial_u m_B = \frac{1}{4} \partial_u D^A D^B C_{AB} - \frac{1}{8} N_{AB} N^{AB} - 4\pi G\, T_{uu}^{M(0)}, \tag{3}$$

where $N_{AB} = \partial_u C_{AB}$ is known as the Bondi news tensor. $T_{uu}^{M(0)}$ is the leading $\frac{1}{r^2}$ coefficient of the matter stress tensor $T_{uu}^{M(0)} = \lim_{r\to\infty} r^2 T_{uu}^M$. It is a function of the radiative data for the matter fields. In the case of the scalar field, $T_{uu}^{M(0)} = \frac{1}{2} \partial_u \mathcal{O} \partial_u \mathcal{O}$. From eqn.(3), we see that $m_B(u, \Omega)$ is determined in terms of radiative data and the integration constant $m_B(u = -\infty, \Omega)$. We will see below that this integration constant is nothing other than the supertranslation charge defined at $\mathcal{I}^+$ [120, 121]. However, we first need to review the phase space of full non-linear general relativity as defined at $\mathcal{I}^+$.

## 2.2 Phase space and conserved charges

It was shown by [121] that the space of free data $C_{AB}(u, \Omega)$ is the radiative phase space of gravity in which the fall-off conditions on the shear field are given by

$$C_{AB}(u, \Omega)|_{u \to \pm\infty} = C_{AB}^{(0)\,\pm}(\Omega) + O\left(\frac{1}{|u|^\delta}\right). \tag{4}$$

It was also shown in [121] that the Poisson bracket structure of the radiative data at $\mathcal{I}^+$ is given by,

$$\{N_{AB}(u, \Omega), C_{MN}(u', \Omega')\} = -16\pi G \delta(u - u') \frac{1}{\sqrt{\gamma}} \delta^2(\Omega - \Omega') \left[\gamma_{A(M}\gamma_{N)B} - \frac{1}{2}\gamma_{AB}\gamma_{MN}\right]. \tag{5}$$

We now return to the integration constant for $m_B$. The initial data for (3) is just the value of $m_B$ at $u = -\infty$ and at each point on the sphere, and gives rise to an infinity of supertranslation charges acting on this phase space. This data is conveniently represented after smearing it with a spherical harmonic on $S^2$

$$\mathcal{Q}_{\ell,m} = \frac{1}{4\pi G} \int \sqrt{\gamma}\, d^2\Omega\, m_B(u = -\infty, \Omega) Y_{\ell,m}(\Omega). \tag{6}$$

These are called the *supertranslation charges* for reasons we explain below. The charge with $\ell = m = 0$ is the Bondi mass at $u \to -\infty$, and it was shown in [122] that this coincides with the standard ADM Hamiltonian [123, 124]. We will treat this charge separately in the discussion below.

In the absence of massive particles (that reach time-like infinity and hence induce non-trivial excitations at $u = +\infty$), we can use eqn.(3) to rewrite the infinity of supertranslation charges as an integrated flux over $\mathcal{I}^+$

$$m_B(u = -\infty, \Omega) = -\frac{1}{4} \int_{-\infty}^{\infty} du\, [-D^A D^B N_{AB} + \frac{1}{2} N_{AB} N^{AB} + 16\pi G\, T_{uu}^{M(0)}]. \tag{7}$$

The supertranslation charges can then be written as a sum of two terms

$$\begin{aligned}
\mathcal{Q}_{\ell,m}^{\text{soft}} &= \frac{1}{16\pi G} \int_{-\infty}^{\infty} du\, d^2\Omega\, \sqrt{\gamma}\, Y_{\ell,m}(\Omega)[\, D^A D^B N_{AB}\,], \\
\mathcal{Q}_{\ell,m}^{\text{hard}} &= -\frac{1}{16\pi G} \int_{-\infty}^{\infty} du\, d^2\Omega \sqrt{\gamma}\, Y_{\ell,m}(\Omega)\,[\frac{1}{2} N_{AB}\, N^{AB} + 16\pi G T_{uu}^{M(0)}].
\end{aligned} \tag{8}$$

The soft charge is generated by "zero mode" of the news tensor given by,

$$\lim_{\omega \to 0} \int_{-\infty}^{\infty} du\, e^{-i\omega u} N_{AB}(u, \Omega) =: \lim_{\omega \to 0} \tilde{N}_{AB}(\omega,\, \Omega), \tag{9}$$

and the hard charge depends on the total (gravitational and matter) stress tensor at $\mathcal{I}^+$.

The action of $\mathcal{Q}_{\ell,m}$ on the phase space of the gravitational radiative data is defined through the following Poisson brackets [125]

$$\{C_{MN}(u,\Omega), \mathcal{Q}_{\ell,m}\} = Y_{\ell,m}(\Omega)\partial_u C_{MN}(u,\Omega) - 2(D_M D_N Y_{\ell,m}(\Omega))^{\text{TF}}. \tag{10}$$

The superscript TF stands for the trace-free component of the tensor. This is precisely the BMS supertranslation symmetry [126] acting on $\mathcal{I}^+$. Apart from the use of the Poisson brackets (5), to obtain the correct inhomogeneous term above, requires a careful consideration of boundary conditions that place a constraint on the integrated news as explained in [114].

So we see that parameterizing the space of all asymptotically flat space-times in terms of the radiative data at $\mathcal{I}^+$ reveals a remarkably simple structure where the phase space of the full non-linear theory is generated by free fields.

The quantization of such a theory was formulated and developed in a series of papers by Ashtekar [112, 120, 125], and is known as the asymptotic quantization program. The elementary operators in this approach are given by the news operators $N_{AB}$.[2] The commutation relations in the quantum theory follow from the Poisson brackets above and are given by,

$$[\, N_{AB}(u,\Omega), N_{CD}(u',\Omega')\,] = i16\pi G\partial_u \delta(u-u')\frac{1}{\sqrt{\gamma}}\delta^2(\Omega-\Omega')\,[\gamma_{A(C}\,\gamma_{D)B} - \frac{1}{2}\gamma_{AB}\gamma_{CD}\,]. \tag{11}$$

One can also define the algebra generated by the shear operators and the zero mode of the news $N_{AB}^{(0)}(\Omega') = \int du\, N_{AB}(u,\,\Omega')$ as [109, 114, 125]

$$[C_{AB}(u,\Omega), C_{CD}(u',\Omega')] = -i8\pi G\Theta(u-u')\frac{1}{\sqrt{\gamma}}\delta^2(\Omega-\Omega')\,[\gamma_{A(C}\,\gamma_{D)B} - \frac{1}{2}\gamma_{AB}\gamma_{CD}\,], \tag{12}$$

where $\Theta(x) = \text{sign}(x)$.

In [125] the algebra generated by the news operators was used to define the Hilbert space of the theory. This was done by first splitting the news operator in terms of creation and annihilation operator defined with respect to $u$. This construction involved several subtleties. It was shown in [125] that the generic Fock states constructed in this manner had divergent norm unless $\int_{-\infty}^{\infty} du\, N_{AB}(u,\,\Omega) = 0\ \forall\ \Omega$. That is, such Fock spaces were constructed only out of those news operators which had no "soft" (i.e. zero) mode. This rather severe and unphysical restriction in quantum theory was overcome by defining the a complete Hilbert space as direct sum over Fock spaces, each of which is defined with respect to a vacuum that contain non-trivial soft modes of the news. We summarize the results of this construction below. We do not detail all the technical subtleties involved in the construction, and more details can be found in a recent review [109].

## 2.3   The Hilbert space of the theory

Decomposing the news tensor and also the massless fields into their positive and negative frequencies via

$$\tilde{N}_{AB}^{\pm}(\omega,\Omega) = \int du e^{\pm i\omega u} N_{AB}(u,\Omega),$$

$$\tilde{O}^{\pm}(\omega,\Omega) = \int du e^{\pm i\omega u} O(u,\Omega), \tag{13}$$

---

[2]We use the same notation for the classical news tensor and the corresponding operator.

defines "creation" and "annihilation" operators which can be used to generate a Fock space. However, to specify the vacuum for this Fock space, we additionally need to specify the action of the zero-mode of the news $N_{AB}^{(0)}(\Omega) = \int_{-\infty}^{\infty} du N_{AB}(u, \Omega)$ on the vacuum.

So a complete specification of the vacuum is obtained not just by demanding that it is annihilated by all the positive-frequency modes of the news, but additionally by specifying its eigenvalue under *all* the supertranslation charges for $\ell > 0$,

$$\mathcal{Q}_{\ell,m}|\{s\}\rangle = s_{\ell,m}|\{s\}\rangle. \tag{14}$$

This eigenvalue is also the eigenvalue of the "soft" part of the supertranslation charge, since the "hard" part annihilates all the vacua. So, the set of all possible vacua are given by specifying a (countably) infinite set of real numbers $s_{\ell,m}$. We will choose these states to be normalized according to

$$\langle\{s\}|\{s'\}\rangle = \prod_{\ell,m} \delta(s_{\ell,m} - s'_{\ell,m}). \tag{15}$$

On top of *each* such vacuum, one can construct a Fock space comprising the states

$$\mathcal{H}_{\{s\}} = \text{span of}\{N(f_1)N(f_2)\ldots N(f_n)O(h_1)\ldots O(h_m)\}|\{s\}\rangle, \tag{16}$$

where $f_1^{AB}(u, \Omega)\ldots f_n^{AB}(u, \Omega)$ and $h_1(u, \Omega)\ldots h_m(u, \Omega)$ are test functions on $\mathbf{R} \times S^2$ and

$$N(f_i) \equiv \int \sqrt{\gamma} N_{AB}(u, \Omega) f_i^{AB}(u, \Omega) du d^2\Omega,$$
$$O(h_i) \equiv \int \sqrt{\gamma} O(u, \Omega) h_i(u, \Omega) du d^2\Omega. \tag{17}$$

Each such space gives an irreducible representation of the algebra of news operators and of the massless matter fields. But the full Hilbert space is obtained by taking the *direct sum* of all of these Hilbert spaces

$$\mathcal{H} = \bigoplus_{\{s\}} \mathcal{H}_{\{s\}}, \tag{18}$$

where the sum is over all possible values of all soft charges.

Some remarks are in order.

1. In the semiclassical theory, the action of the shear operators does not generate any states beyond $\mathcal{H}$ and moreover $\mathcal{H}$ turns into an irreducible representation when these operators are included in the algebra. This is because the constant shear mode $C_{AB}^{(0)}(\Omega)$, which is independent of $u$, is conjugate to the soft charge [114]. Its action on one of the vacua is

$$e^{-\frac{i}{2} \int \sqrt{\gamma} F^{AB}(\Omega) C_{AB}^{(0)}(\Omega) d^2\Omega} |\{s\}\rangle = |\{s'\}\rangle, \tag{19}$$

where

$$s'_{\ell,m} = s_{\ell,m} + \int \sqrt{\gamma} F^{AB}(\Omega)\left(D_A D_B - \frac{1}{2}\ell(\ell+1)\gamma_{AB}\right) Y_{\ell,m}(\Omega) d^2\Omega. \tag{20}$$

2. Within the semiclassical theory, acting with the Bondi mass aspect on the vacuum also does not generate any new states beyond $\mathcal{H}$ since it is related by the constraint (3) to the news operators and the supertranslation charges. So we have not displayed this action separately either in (16).

3. To obtain an IR-finite S-matrix, it is convenient to dress the matter fields with soft gravitons according to the Faddeev-Kulish [127] prescription. (This also corresponds to the fact that the massless matter field develops log-tails at $\mathcal{I}^+$ due to the long range gravitational field.) In our description, the matter fields are bare. But the soft-charges of the vacua we use can be interpreted as arising from a background cloud of soft gravitons. The relationship between these two descriptions is explored in [116, 128, 129].

4. The inclusion of massive particles does not invalidate the construction above for massless particles. Massive particles that reach time-like infinity are dressed by Coulombic modes of the gravitational field. Thus in the semiclassical theory, the Hilbert space of massive particles is independent of the Hilbert space of radiative modes of gravitational field [130].

5. Although we have labelled the vacua only by their supertranslation soft charges, recent studies in asymptotic symmetries indicate that the so-called superrotation soft charges commute with the supertranslation soft charges [131, 132], and this may further refine the vacuum structure of the theory. As we explain below, our analysis will not be affected by such refinements, and so, for the sake of simplicity in presentation, we restrict our vacuum-labels to supertranslations.

To avoid any confusion due to the points above we make the following definition that we will use consistently in this paper.

**Definition 1.** *The Hilbert space of massless particles, $\mathcal{H}$, refers to the space obtained by starting with all possible vacua, exciting each vacuum with operators on $\mathcal{I}^+$ and then taking the span of all states so obtained.*

In the semiclassical theory, this leads to the space described by Eqn. (18) but the definition above holds generally.

## 2.4 Algebras in the neighbourhood of a cut

We have seen that there is a nice construction of the algebra of diffeomorphism invariant operators on $\mathcal{I}^+$, and the corresponding Hilbert space at $\mathcal{I}^+$. However this algebra, $\mathcal{A}(\mathcal{I}^+)$, involves the news operators at all points of $\mathcal{I}^+$. Since we wish to understand how much information is available near a given cut of null infinity, we now define the notion of an algebra in the neighbourhood of a cut that will play a key role in our analysis below.

**Definition 2.** *The algebra associated with an $\epsilon$-neighbourhood of a cut $u_0$ is denoted by $\mathcal{A}_{u_0,\epsilon}$ and consists of all possible functions of asymptotic operators with a u-coordinate lying in $(u_0, u_0 + \epsilon)$.*

To avoid any confusion, we now elaborate on this definition. Allowing $u \in (u_0, u_0 + \epsilon)$ and $\Omega$ to range over the entire celestial sphere, we take $\mathcal{A}_{u_0,\epsilon}$ to comprise the set of all possible functions of $m_B(u, \Omega)$, $C_{AB}(u, \Omega)$, and the massless matter fields collectively denoted as $O(u, \Omega)$. For instance some of the lowest order polynomials that are elements of $\mathcal{A}_{u_0,\epsilon}$ are

$$\mathcal{A}_{u_0,\epsilon} = \{m_B(u_1, \Omega), C_{AB}(u_1, \Omega_1), O(u_1, \Omega_1), m_B(u_1, \Omega_1)C_{AB}(u_2, \Omega_2), \\ m_B(u_1, \Omega_1)O(u_2, \Omega_2), C_{AB}(u_1, \Omega_1)O(u_2, \Omega_2), O(u_1, \Omega_1)O(u_2, \Omega_2)\ldots\}, \tag{21}$$

where $u_i \in (u_0, u_0 + \epsilon)$.

In a rigorous treatment, we would consider only bounded functions of the elementary operators rather than all functions. While we do not see any obstacle to reformulating our results in that language, we do not adopt that approach here for the sake of simplicity.

We would like to emphasize two points.

- Apart from the massless matter fields, $\mathcal{A}_{u_0,\epsilon}$ is generated by the shear operators as well as the Bondi mass aspect. These are simply the components of the metric in the Bondi gauge. In the asymptotic quantization program, the Bondi mass aspect is related by constraints to the integrated square of the news. However, this does *not* mean that it ceases to be an observable at $u$. For instance, if one were to compute the components of any other composite observable of the metric such as the Riemann tensor in the vicinity of $u$, *both* the Bondi mass and the shear would enter this computation.

- As is standard in the analysis of algebras in quantum field theory, our algebra includes polynomials and other functions constructed out of the elementary operators as well as their spectral projections. Indeed, the spectral projections can themselves be obtained just as limits of functions of the operators.

Finally an important special case of the definition above is the algebra obtained near the boundary of null infinity.

**Definition 3.** *The algebra near the past boundary of future null infinity, $\mathcal{A}_{-\infty,\epsilon}$ is the set of all functions of operators on $\mathcal{I}^+$ with u-coordinate in $(-\infty, -\frac{1}{\epsilon})$.*

The algebra near the future boundary of past null infinity is defined similarly. To elaborate on the definition, $\mathcal{A}_{-\infty,\epsilon}$ comprises all functions of $C_{AB}(u,\Omega), O(u,\Omega), m_{\mathrm{B}}(u,\Omega)$, as shown in (21) except that the range of $u$ is $u \in (-\infty, -\frac{1}{\epsilon})$.

## 3 Holographic storage of quantum information at null infinity

In this section we will state and prove the two main results of our paper. While reading this section, we would like to remind the reader to keep the philosophy of this paper in mind. We will assume, based on physical justifications, that some properties of the semiclassical theory reviewed above can be extrapolated to the full theory of quantum gravity. We will first make a weaker extrapolation, which only relies on low-energy physics and derive result (1) below. Since this result relies on very weak assumptions, it is very robust. We will then explore a stronger extrapolation, which allows for the stronger result (2).

### 3.1 Information at the past of future null infinity

Our first main result is as follows.

**Result 1.** *Any two distinct states in the Hilbert space of massless particles can be distinguished just by observables in an infinitesimal neighbourhood of $\mathcal{I}^+_-$.*

What we need to prove is as follows. Consider any two states, $|\Psi_1\rangle$ and $|\Psi_2\rangle$ and say that there exists some operator $\mathcal{H} \to \mathcal{H}$ that takes on different values in these states,

$$\langle \Psi_1|A|\Psi_1\rangle \neq \langle \Psi_2|A|\Psi_2\rangle, \qquad A \in \mathcal{A}(I^+). \tag{22}$$

Then we need to find an element of the algebra of operators localized near the past of future null infinity — which we termed $\mathcal{A}_{-\infty,\epsilon}$ above — that can also distinguish between these two states.

The first important assumption we will need to prove Result (1) is as follows.

**Assumption 1.1.** *The vacua in the full theory of quantum gravity can be completely identified by the values of operators near $\mathcal{I}^+_-$*

As explained in section 2, this assumption is consistent with our current understanding of the vacuum structure of quantum gravity in asymptotically flat space. The vacua are ground states of the ADM Hamiltonian, which is defined at $\mathcal{I}^+_-$, but they carry further charges. In the analysis below, to lighten the notation, we will assume that these additional charges are only the supertranslations, which are manifestly defined at $\mathcal{I}^+_-$. As mentioned in section 2, there are indications that additional charges may be required to distinguish between the vacua. But this can be accommodated within Assumption (1.1) provided these additional charges can all be defined at $\mathcal{I}^+_-$ (as is the case for the so-called super rotation charges).

Assumption (1.1) is a good assumption because it just pertains to the *low energy* structure of the full theory of quantum gravity. We believe that this low energy structure is captured by effective field theory, and while effective field theory may be insufficient to capture the fine details of the ultraviolet Hilbert space, it does correctly capture the vacuum structure.

We now note a second property of the semiclassical theory: the algebra obtained near the past boundary of $\mathcal{I}^+$ can not only be used to identify the vacua, but also induce transitions between any two vacua. This can be seen as follows. We first recall that the projector onto states of zero ADM mass is an element of this algebra. These states are all labelled by a distinct supertranslation charge and so this projector, $P_0$, can be expanded as

$$P_0 = \int \left( \prod_{\ell,m} ds_{\ell,m} \right) |\{s\}\rangle\langle\{s\}| \in \mathcal{A}_{-\infty,\epsilon}. \tag{23}$$

Now we can perform a spectral decomposition for each supertranslation charge[3]

$$\mathcal{Q}_{\ell,m} = \int ds\, s\, \mathcal{P}_{\ell,m}[s]. \tag{24}$$

Now, this supertranslation charge includes both hard and soft parts but by multiplying the projector onto the space of vacua with an infinite product of $\mathcal{P}_{\ell,m}[s]$ we can select a *specific soft vacuum*

$$P_0 \prod_{\ell,m} \mathcal{P}_{\ell,m}[s_{\ell,m}] = |\{s\}\rangle\langle\{s\}| \in \mathcal{A}_{-\infty,\epsilon}. \tag{25}$$

Second consider starting with a particular soft vacuum, acting with a smeared shear operator and then projecting back onto the space of all vacua. It is easy to see that this leads to another vacuum with a different value of the soft charges.

$$P_0 e^{-\frac{i}{2}\int_{-\infty}^{-\frac{1}{\epsilon}} \sqrt{\gamma} C_{AB}(u,\Omega) G^{AB}(u,\Omega) d^2\Omega} |\{s\}\rangle = |\{s'\}\rangle, \tag{26}$$

where we see from (20) that

$$s'_{\ell,m} = s_{\ell,m} + \int_{-\infty}^{-\frac{1}{\epsilon}} du\, d^2\Omega\, \sqrt{\gamma} G^{AB}(u,\Omega) \left( D_A D_B - \frac{1}{2}\ell(\ell+1)\gamma_{AB} \right) Y_{\ell,m}(\Omega). \tag{27}$$

Since we can choose $G$ to be arbitrary, we can attain any value of $s'_{\ell,m}$ starting with a given value of $s_{\ell,m}$.

Therefore, using operator from the algebra, one can not only select a particular vacuum but also cause transitions to any other vacuum.

$$T_{\{s\},\{s'\}} = |\{s\}\rangle\langle\{s'\}| \in \mathcal{A}_{-\infty,\epsilon}. \tag{28}$$

---

[3]The vacuum projector can be constructed explicitly as a limit of a bounded functions on $\mathcal{A}_{-\infty,\epsilon}$ through $P_0 = \lim_{\alpha\to\infty} e^{-\alpha M(-\infty)}$. The operator, $\mathcal{P}_{\ell,m}[s]$, which we use for ease of notation, selects a delta-function normalized state so it is not bounded. But the spectral projector onto any range of values of $\mathcal{Q}_{\ell,m}$ can also be constructed as a limit of bounded functions on $\mathcal{A}_{-\infty,\epsilon}$: $\int_s^{s'} \mathcal{P}_{\ell,m}[x] dx = \lim_{T\to\infty} \frac{1}{2\pi} \int_{-T}^T e^{i\theta \mathcal{Q}_{\ell,m}} \frac{e^{-i\theta s} - e^{-i\theta s'}}{i\theta} d\theta$.

So, in the canonical theory, any operator that maps the space of vacua back to itself can be written as a linear combination of the transition operators above, and this proves the lemma.

The argument above crucially used the low-energy behaviour of the news-shear commutator in equation (27). We expect this to be robust since the UV-theory should not change low-energy physics. Moreover, the argument above only uses *bounded operators* at each step. Nevertheless, it may be the case that the detailed form of this commutator is modified in the full theory of quantum gravity, so that (27) receives corrections. However, we will need to assume that in the full theory at least the following property holds.

**Assumption 1.2.** *All operators that map the space of vacua back to itself are contained in $\mathcal{A}_{-\infty,\epsilon}$.*

As argued above, this assertion is true in the canonical theory. It appears reasonable to assume that the full theory of quantum gravity will share this property since it again only involves *low energy physics*.

Finally, we need a third physical assumption.

**Assumption 1.3.** *The spectrum of the Hamiltonian of the full theory of quantum gravity is bounded below.*

This seems, to us, to be a very natural assumption, and so we do not justify it any further. We merely note that the assumption above is weaker than any of the commonly used energy conditions since it says nothing about the local positivity of energy but merely about its global positivity. For convenience, we will choose this lower bound to be 0 and simply assume, below, that the energy eigenvalues are positive.

The assumption (1.3) immediately leads to the following Lemma.

**Lemma 1.** *The Hilbert space $\mathcal{H}$ can also be generated by starting with all possible vacua and acting with operators from an infinitesimal neighbourhood of the the past of future null infinity.*

First consider the sector built on top of a particular vacuum as displayed in Eqn. (16) by smeared news and matter operators. What we need to prove is that all these states can be generated just by acting with operators near the past of future null infinity

$$\mathcal{H}_{\{s\}} = \text{span of } \{N(\widetilde{f}_1)N(\widetilde{f}_2)\ldots N(\widetilde{f}_n)O(\widetilde{h}_1)\ldots O(\widetilde{h}_m)\}|\{s\}\rangle, \tag{29}$$

where the notation is the same as (17) except that $\widetilde{f}_i$ and and $\widetilde{h}_i$ have support *only* for $u \in (-\infty, -\frac{1}{\epsilon})$. Note that this support is very different from the support of the functions $f_i$ and $h_i$ in equation (16) that could be the entire real line.

We will prove the statement via contradiction. Imagine that there exists a state, $|\Psi_\perp\rangle$, that belongs to the Hilbert space but is orthogonal to all states of the form above. This implies that whenever $u_i \in (-\infty, -\frac{1}{\epsilon})$, the following correlator vanishes.

$$\kappa(u_i) = \langle\Psi_\perp|N_{A_1B_1}(u_1,\Omega_1)\ldots N_{A_nB_n}(u_n,\Omega_n)O(u_{n+1},\Omega_{n+1})\ldots O(u_{n+m},\Omega_{n+m})|\{s\}\rangle = 0. \tag{30}$$

We may now insert a complete set of eigenstates of the full Hamiltonian to evaluate the correlator above.

$$\kappa(u_i) = \sum_{E_i} \langle\Psi_\perp|E_1\rangle\langle E_1|N_{A_1B_1}(0,\Omega_1)|E_2\rangle\ldots\langle E_{n+m}|O(0,\Omega_{n+m})|\{s\}\rangle e^{i\sum_{i=1}^{n+m} E_i z_i}, \tag{31}$$

where the variables $z_i$ are defined as

$$z_1 = u_1; \quad z_2 = u_2 - u_1; \quad \ldots \quad z_{n+m} = u_{n+m} - u_{n+m-1}. \tag{32}$$

As a function of these variables, and as a result of our assumption about the *positivity* of the $E_i$ above, we find that $\kappa$ is analytic when we extend the $z_i$ to the *upper half plane*.

Now, by the *edge of the wedge* theorem [133], if $\kappa$ vanishes for all $u_i \in (-\infty, -\frac{1}{\epsilon})$, it must vanish for all real $u_i$. But this is impossible since, by assumption, $|\Psi_\perp\rangle$ is itself generated by acting with news operators on the vacuum. Therefore, we have reached a contradiction with our initial assumption. So $|\Psi_\perp\rangle$ cannot exist. This proves the lemma. ∎

We would like to make two short remarks. Although we have focused on a neighbourhood near $\mathcal{I}_-^+$, the same argument above shows that any sector of the Hilbert space can be generated by acting with operators from *any* infinitesimal neighbourhood of future null infinity. Second the argument above shows that Assumption 1.2 can also be phrased as an assumption about $\mathcal{A}(\mathcal{I}^+)$ rather than an assumption about $\mathcal{A}_{-\infty,\epsilon}$.

**Proof of result** (1)  We now move to the proof of result (1). Let us expand the operator that distinguishes between the two states as

$$A = \sum_{s,s',n,m} c(n,m,s,s')|n_{\{s\}}\rangle\langle m_{\{s'\}}|, \tag{33}$$

where the state $|n_{\{s\}}\rangle$ belongs to the sector of the Hilbert space built on top of the soft vacuum $|\{s\}\rangle$ i.e. $|n_{\{s\}}\rangle \in \mathcal{H}_{\{s\}}$, the state $|m_{\{s'\}}\rangle$ belongs to the sector of the Hilbert space built on top of the soft vacuum $|\{s'\}\rangle$ i.e. $|m_{\{s'\}}\rangle \in \mathcal{H}_{\{s'\}}$, and the coefficients $c(n,m,s,s')$ are $c$-numbers.

But, by the result above, we can write

$$|n_{\{s\}}\rangle = X_n|\{s\}\rangle; \qquad |m_{\{s'\}}\rangle = X_m|\{s'\}\rangle, \tag{34}$$

where the operators $X_n, X_m$ both belong to the algebra that lives near the past boundary of future null infinity: $X_n, X_m \in \mathcal{A}_{-\infty,\epsilon}$. Combining this with the assumption about operators in the space of vacua above, we find that we can write the entire operator as

$$A = \sum c(n,m,s,s')X_n T_{\{s\},\{s'\}} X_m^\dagger. \tag{35}$$

Since every operator on the right hand side of (35) belongs to the algebra $\mathcal{A}_{-\infty,\epsilon}$, and since the algebra is closed under products and linear combinations by construction, we find that $A \in \mathcal{A}_{-\infty,\epsilon}$.

Therefore the states, $|\Psi_1\rangle$ and $|\Psi_2\rangle$ can be distinguished just by elements of $\mathcal{A}_{-\infty,\epsilon}$, which only comprises operators in the neighbourhood of spatial infinity. ∎

## 3.2 The nested structure of information on cuts of null infinity

We now turn to our second main result. The second result states that if information is available at any cut of $\mathcal{I}^+$, it is also available in the *past* of that cut. However, the converse statement is not true.

**Result 2.** *Any two states that are distinguishable by operators in $\mathcal{A}_{u_1,\epsilon}$ can be distinguished by operators in $\mathcal{A}_{u_2,\epsilon}$ for any $u_2 < u_1$.*

The careful reader will note that result (2) is stronger than result (1) and, in fact, result (1) may be understood as a special case of result (2) when we take $u_2 \to -\infty$. But, as we will see below, to prove result (2) also requires stronger assumptions about the UV-complete theory.

To motivate this result, we first need to massage the commutators and constraints reviewed in section 2. First by integrating the constraint equation for the Bondi mass aspect, given in (3), over the sphere we find that the Bondi mass, $M(u)$ defined as

$$M(u) = \int \sqrt{\gamma} m_\text{B}(u,\Omega) d^2\Omega, \tag{36}$$

satisfies the constraints

$$\partial_u M(u) = -\int \sqrt{\gamma} d^2\Omega \left[ \frac{1}{8} N_{AB} N^{AB} + 4\pi G T_{uu}^{M(0)} \right].$$  (37)

Using the news-news commutators and the constraint equation given in (3), we see immediately that

$$[\partial_u M(u), C_{AB}(u',\Omega)] = 4\pi G i \partial_{u'} C_{AB}(u',\Omega) \delta(u-u').$$  (38)

The stress-tensor of the matter fields that appears in (3) has simple commutators with the matter field

$$[T_{uu}^{M(0)}(u,\Omega), O(u',\Omega')] = \frac{-i}{\sqrt{\gamma}} \partial_{u'} O(u',\Omega) \delta(u-u') \delta^2(\Omega-\Omega').$$  (39)

Therefore we see that the commutator of the derivative of the Bondi mass with any matter field also has the same form as its commutator with components of the metric

$$[\partial_u M(u), O(u',\Omega)] = 4\pi G i \partial_{u'} O(u',\Omega) \delta(u-u').$$  (40)

Note that no factor of $\gamma$ appears in this expression.

   We now need to set initial conditions to derive the commutator of the Bondi mass with dynamical fields. We assume that, even in the full quantum theory, as $u \to -\infty$, the integrated Bondi mass tends to the canonical Hamiltonian

$$\lim_{u\to-\infty} \frac{1}{4\pi G} M(u) = H.$$  (41)

We expect that the commutator of the Hamiltonian with the metric and matter fields at null infinity simply generates translations along null infinity

$$\begin{aligned} [H, C_{AB}(u,\Omega)] &= -i\partial_u C_{AB}(u,\Omega), \\ [H, O(u,\Omega)] &= -i\partial_u O(u,\Omega). \end{aligned}$$  (42)

   Then using the constraint equation on $M(u)$ above and the commutators of $M(u)$ with the news, this leads to the following commutators of $M$.

$$\begin{aligned} [M(u), C_{AB}(u',\Omega)] &= -4\pi G i \partial_{u'} C_{AB}(u',\Omega) \theta(u'-u), \\ [M(u), O(u',\Omega)] &= -4\pi G i \partial_{u'} O(u',\Omega) \theta(u'-u). \end{aligned}$$  (43)

The commutators above can be simply generalized to *any polynomial* in the metric and matter fields, and have a very simple form. Taking a commutator of any observable at $u'$ with the Bondi mass at $u$ is just like taking a $u'$-derivative of the observable if $u' > u$; otherwise the commutator vanishes.

   The commutators (43) are exact in the full nonlinear Einstein theory. To prove our second result above, we will need to make the following assumption.

**Assumption 2.1.** *In the full theory of quantum gravity, the commutators of the Bondi mass, $M(u)$, with other asymptotic fields (given in (43)) and the evolution equation for the Bondi mass (given in (3) ) are exact up to possible corrections by local operators in the algebra at $u$.*

   This may seem like a strong assumption, but it can be demonstrated to all orders in effective field theory. The reason for this is that since the commutators in (43) are derived at null infinity, they depend just on the *weak-field* structure of the theory and therefore only on those terms in the action that are *quadratic* in the fields. So even if one adds an infinite number of higher-derivative interactions to the Lagrangian, provided all of these terms modify only the nonlinear

interaction-terms, they *all* become unimportant at null infinity. Within effective field theory, this captures all possible terms that can appear in the effective action. Now, the assumption above may fail nonperturbatively. But, in this case, the results that we derive below would still be valid at all orders in perturbation theory. We once again recall that from our perspective, the infrared effects do not alter the commutator algebra and only affect the vacuum structure of quantum gravity.

**Proof of result** (2)    Subject to the assumption above, result (2) now follows in a single step from our analysis. The commutators (43) lead to a differential equation for the dynamical fields in the theory. Consider two points $u_0, u_0' \in (u_2, u_2 + \epsilon)$ with $u_0' > u_0$. Since the algebra in the vicinity of the cut at $u_2$ includes both $M(u_0)$ and the matter and metric fields at $u_0'$, we can use these to set the initial conditions for the differential equation. This differential equation has a *unique* solution as we evolve towards the future of null infinity. Explicitly, we have

$$
\begin{aligned}
C_{AB}(u_0' + U, \Omega) &= e^{\frac{iM(u_0)}{4\pi G}U} C_{AB}(u_0', \Omega) e^{\frac{-iM(u_0)}{4\pi G}U}; \\
O(u_0' + U, \Omega) &= e^{\frac{iM(u_0)}{4\pi G}U} O(u_0', \Omega) e^{\frac{-iM(u_0)}{4\pi G}U};
\end{aligned}
\tag{44}
$$

for any $U > 0$.[4] Once we have the operator values for all the matter fields, we may obtain the value of $M(u_0 + U)$ by solving the constraint equation (3). By taking $U = u_1 - u_2$ we obtain all the operators in the algebra obtained near the cut $u_1$. ∎

Note that this process is *not* reversible and the equation (44) does not hold for $U < 0$ because the differential equation ceases to be valid in that domain due to the $\theta$-function in (43). So, the structure of future null infinity is asymmetric in its information content. As we move towards the future, we lose information.

Of course, an analogous result holds at past null infinity. There, the information in any cut of past null infinity is also contained in any cut to the future.

### 3.3   Some subtleties and explanations

The results above are very surprising. They indicate that even in asymptotically flat space, all the information that is available within gravitational radiation and the radiation of other massless particles can be obtained in the vicinity of spatial infinity *without* waiting for this radiation to physically reach null infinity. Since this is very different from the manner in which quantum information is stored in local quantum field theories, we now discuss some subtleties and provide explanations for some potentially confusing points.

**The Importance of Gravity.**

The result above crucially requires gravity. This is an elementary consistency check since it is clear that no result of the form of result (1) or result (2) can hold in ordinary local quantum field theories. In a local quantum field theory with massless particles, one should be free to specify the quantum wavefunction separately on different parts of $\mathcal{I}^+$. So in such theories it clearly cannot be the case that all information about the wavefunction is already contained in a vicinity of $\mathcal{I}^+_-$. The results above are also false in nongravitational gauge theories.

From a technical perspective, this happens because Assumption 1.1 is *false* in any nongravitational theory. In no theory, except for quantum gravity can one project onto the vacuum purely using operators near $\mathcal{I}^+_-$. In a nongravitational gauge theory while one can project onto states of zero charge, there is an *infinity* of such states. So asymptotic operators are insufficient to pinpoint a vacuum and then the rest of the argument that leads to result (1) falls apart.

---

[4]We start evolving the fields from $u_0' > u_0$ rather than $u_0$ to avoid any subtleties with the the value of the theta function when its argument is exactly 0.

The argument that leads to result (2) also cannot be generalized to a nongravitational setting since there is no analogue of the Bondi mass at a cut in a nongravitational theory that can be used to evolve operators into the future,

Indeed in nongravitational theories it is easy to construct a counterexample to (1) and (2). Consider any states $|\Psi\rangle$ and another state $U|\Psi\rangle$ obtained by exciting the original state with a *gauge-invariant* unitary operator from the algebra near the cut at $u = 0$. For instance, in QED we may take

$$U = e^{i \lim_{r\to\infty} r^2 \int \sqrt{\gamma} F_{\mu\nu}(r,u,\Omega) F^{\mu\nu}(r,u,\Omega) f(u,\Omega) d^2\Omega du}, \tag{45}$$

where $f$ smears the operator in a small region near $u = 0$. In the absence of gravity, such an operator commutes with all operators in the algebra near any cut except for the algebra near the cut at $u = 0$. So it is impossible to distinguish $|\Psi\rangle$ and $U|\Psi\rangle$ either at $\mathcal{I}^+_-$ or at any cut at negative $u$.

The results established above show that, in gravity, the constraints are much stronger and contain much more information than they do in any other theory. In fact we explain below why an attempt to "hide" information from spatial infinity using a construction of the form (45) fails in gravity.

**Perturbative verification**

The reader may have found our results somewhat formal. However these results can be verified already in perturbation theory. We will discuss this in detail in forthcoming work [134] but we provide a limited preview of these results here.

Consider a vacuum, $|\Omega\rangle$ formed by taking an arbitrary superposition of the soft vacua detailed above and normalized so that $\langle\Omega|\Omega\rangle = 1$. Now, we excite this vacuum by acting on it with a unitary operator that comprises the news insertion smeared with a function of *compact support* near $u = 0$.

$$|f\rangle = e^{i\lambda \int du d^2\Omega \sqrt{\gamma} N_{AB}(u,\Omega) f^{AB}(u,\Omega)} |\Omega\rangle. \tag{46}$$

The challenge is to back-calculate the function $f^{AB}$ using observations only in the vicinity of $u = -\infty$. The construction above is just like (45) but in gravity, unlike QED, the challenge can be met.

A simple calculation shows that this can be done by considering the *two-point* function of the Bondi mass at $\mathcal{I}^+_-$ and news operator insertions in the interval $(-\infty, -\frac{1}{\epsilon})$.

$$\langle f | M(-\infty) N_{CD}(u,\Omega') | f \rangle = \lambda \int dx \, 16G \frac{f^{AB}(x,\Omega')}{(x-u-i\epsilon)^3} [\, \gamma_{A(M}\gamma_{N)B} - \frac{1}{2}\gamma_{AB}\gamma_{MN} \,] + O(\lambda^2). \tag{47}$$

Since the function on the right hand side is analytic when $u$ is extended in the upper half plane given its value for $u \in (-\infty, -\frac{1}{\epsilon})$ we can reconstruct $f^{AB}$. A similar calculation allows one to extract $f^{AB}$ from the neighbourhood of any cut for negative $u$.

This calculation also explains why we need an *infinitesimal interval* rather than a cut. Using the value of the two point correlator, (47), for only for a fixed value of $u$ it is not possible to reconstruct $f^{AB}$. It may still be possible to reconstruct $f^{AB}$ by using correlators of arbitrarily complicated complicated operators at a single value of $u$. But using a small interval obviates the need for such complicated correlators and allows a perturbative examination of how holographic information is stored. As explained above, this computation crucially relies on gravity and on the nonvanishing commutators of the Bondi mass with other operators.

**The Importance of Quantum Mechanics.**

The discussion above makes it clear that the results (1) and (2) do *not* hold in the classical theory.

In fact the states discussed in the previous paragraph already provide a counterexample in the classical theory. Consider a time-reversed Vaidya solution that consists of an expanding shell that crosses $\mathcal{I}^+$ at some value of retarded time. We may consider another classical solution comprising a shell that crosses $\mathcal{I}^+$ at a *different* value of the retarded time. These solutions provide the classical description of a state obtained by exciting the vacuum with a news operator in the vicinity of a cut and they are characterized by the location of this cut.

But, in the classical theory in the vicinity of spatial infinity, except for the total mass of the solution we cannot determine any of its other details or the retarded time at which it will cross $\mathcal{I}^+$.

It is not surprising that there is no classical analogue of our quantum mechanical results. Classically, the constraints can only determine the *expectation value* of the Bondi mass, $\langle M(u) \rangle$. However, to obtain information about the state we need the quantum correlators of the Bondi mass with other operators and there is no classical analogue of this quantity.

The non-uniqueness of classical boundary data near spatial infinity is well known in the literature [135] and a nice discussion of the contrast between the amount of information available classically and quantum mechanically is also given in [136].

**Nongravitational limit.**

We have emphasized that in quantum gravity, information about the state is already contained in the vicinity of spatial infinity and moreover information available at a future cut is already available in all cuts to its past.

However, it is important to also understand the settings for which this result is not relevant: *in the limit where we take $M_{pl} \to \infty$ and ignore the information in gravitational correlators, we recover the usual picture of local quantum field theory where information is stored locally rather than holographically*. When such a decoupling limit is possible, results (1) and (2) remain true but may not be relevant from a practical perspective.

All quantum-information experiments that are feasible with current technology fall into the category above. For instance, if one is given a sealed box of qubits, in the real world, it is not practical to read off the qubits just by making measurements of the quantum fluctuations of the metric around the box, and the only practical possibility is to open the box and directly examine the qubits.

This is an obvious point but nevertheless we urge the reader to keep it in mind. Our everyday intuition about the localization of quantum information is built by our experiences in a regime where $M_{\mathrm{pl}}$ is very large compared to other energy scales. Results (1) and (2) are in conflict with this intuition because they are relevant in a regime where effects suppressed by $M_{\mathrm{pl}}$ are important.

# 4 Relevance for the information paradox

A significant amount of attention has been devoted to the question of "how information emerges from a black hole." This discussion is strongly predicated on our everyday intuition for quantum information. In both local quantum field theory and classical physics, if we consider some object that forms and breaks up, the only way to recover information about the object is by slowly collecting its pieces as they emerge.

However, the analysis of the previous section suggests that this is entirely the wrong question in quantum gravity. Since an infinitesimal neighbourhood of $\mathcal{I}^-_-$ already has all information about massless particles, nothing is achieved by waiting for the black hole radiation to physically arrive at null infinity. This is a manifestation of the fact that the information is already outside in quantum gravity and never "goes in"!

This can be formalized in the following two results, which follow immediately from our previous results.[5]

**Result 3.** *The fine-grained von Neumann entropy of the segment* $(-\infty, u_0)$ *of* $\mathcal{I}^+$ *is independent of* $u_0$ *for any pure or mixed state on* $\mathcal{H}$.

This result requires only result (1) as we show below. If we also assume result (2), we find a stronger result

**Result 4.** *The fine-grained von Neumann entropy of the segment* $(u_1, u_2)$ *of* $\mathcal{I}^+$ *with* $u_2 > u_1$ *is independent of* $u_2$ *for any state.*

We first establish these results and then discuss their interpretation.

**Proof of result 3**

First, we review how the von Neumann entropy of the state at future null infinity up to a cut at $u_0$ is defined.

The first step is to consider the algebra, $\mathcal{B}_{-\infty, u_0}$, formed by considering all possible functions of operators on $\mathcal{I}^+$ that lie in $(-\infty, u_0)$. The definition is precisely analogous to the definition of the algebras in the vicinity of a cut that we have considered previously, except that we allow the operators to be localized within a larger interval.

$$\mathcal{B}_{-\infty, u_0} = \{m(u_1, \Omega_1), C_{AB}(u_1, \Omega_1), O(u_1, \Omega_1), m(u_1, \Omega_1)C_{AB}(u_2, \Omega_2), \\ m(u_1, \Omega_1)O(u_2, \Omega_2), C_{AB}(u_1, \Omega_1)O(u_2, \Omega_2)\ldots\}, \tag{48}$$

where $u_i \in (-\infty, u_0)$.

Now consider any density matrix from the Hilbert space $\mathcal{H}$, which we denote by $\sigma$. Recall that by the definition of $\mathcal{H}$ above that the algebra $\mathcal{B}_{-\infty, u_0}$ maps $\mathcal{H}$ back to itself. Now the *reduced* density matrix associated with a segment is defined to be the element of the algebra of the segment $\mathcal{B}_{-\infty, u_0}$ that, when traced with any other observable in the algebra, reproduces the expectation value of the observable given by the density matrix $\sigma$. More precisely, we choose the reduced density matrix of the segment, $\rho_{-\infty, u_0}$, to satisfy

$$\text{Tr}(\rho_{-\infty, u_0} b) = \text{Tr}(\sigma b), \quad \forall\, b \in \mathcal{B}_{-\infty, u_0}, \tag{49}$$

subject to the condition that $\rho_{-\infty, u_0} \in \mathcal{B}_{-\infty, u_0}$. The von Neumann entropy of the segment is now defined as

$$S_{-\infty, u_0} = -\text{Tr}(\rho_{-\infty, u_0} \log(\rho_{-\infty, u_0})). \tag{50}$$

However, in result 3 we proved that *any operator* that mapped $\mathcal{H} \to \mathcal{H}$ could be approximated arbitrarily well by an operator in $\mathcal{A}_{-\infty, \epsilon}$. So $\sigma \in \mathcal{A}_{-\infty, \epsilon}$. Therefore, we can always choose

$$\rho_{-\infty, u_0} = \sigma \in \mathcal{A}_{-\infty, \epsilon}, \tag{51}$$

but this choice is independent of $u_0$. Therefore

$$S_{-\infty, u_0} = -\text{Tr}(\sigma \log(\sigma)), \tag{52}$$

which is manifestly independent of $u_0$! ∎

---

[5]We frame these results in terms of density matrices, traces and and von Neumann entropy since we anticipate that these concepts are more familiar to most readers even if they need to be carefully defined and regulated. For the more rigorously-minded reader, we note that similar results hold for the relative-entropy of two states: it is independent of the upper-bound of the segment of $\mathcal{I}^+$ on which it is evaluated.

**Proof of result 4**

The proof of result 4 is precisely analogous to the proof above and so we only sketch it.

To define the reduced density matrix associated with a segment, we first define the algebra $\mathcal{B}_{u_1,u_2}$ in precisely the same fashion as above. Now consider a density matrix, $\mu$, in the full quantum theory and for the purposes of this result, such a state may have both *both massless and massive* excitations. Then the reduced density matrix we are looking for is defined by the condition

$$\text{Tr}(\rho_{u_1,u_2} b) = \text{Tr}(\mu b), \quad \forall\, b \in \mathcal{B}_{u_1,u_2}, \tag{53}$$

subject to the constraint $\rho_{u_1,u_2} \in \mathcal{B}_{u_1,u_2}$. But since, by result (2) *any* in $\mathcal{B}_{u_1,u_2}$ can be written as an operator in $\mathcal{A}_{u_1,\epsilon}$ we can always choose

$$\rho_{u_1,u_2} \in \mathcal{A}_{u_1,\epsilon}. \tag{54}$$

This choice is manifestly independent of $u_2$ and so the von Neumann entropy of this density matrix is also independent of $u_2$. ∎.

The only reason we separate results (3) and (4) is that they depend, respectively, on result (1) and result (2). So if the stronger assumptions that are required to establish the result (2) fail, the result (3) would still remain valid.

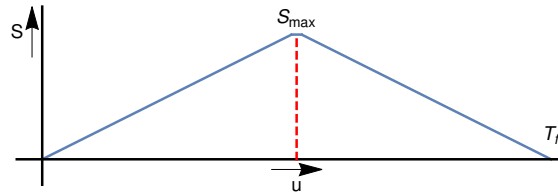

Figure 2: The naive Page curve. If one incorrectly assumes that the Hilbert space factorizes into degrees of freedom outside and inside the black hole, the von Neumann entropy of the radiation that has emerged till retarded time $u$ on $\mathcal{I}^+$ is expected to obey the curve above indicating that "information is gradually returned to the exterior."

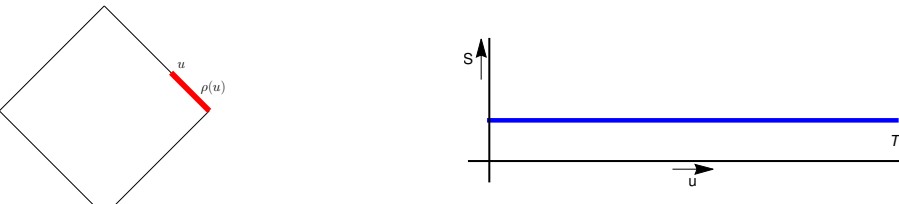

Figure 3: The fine-grained von Neumann entropy (right) of any state of massless excitations reduced on a segment that extends till the cut at $u$ marked on the left figure. The result directly follows from the arguments above. The information is always outside, and so the entropy never goes up or comes down!

While we expect the von Neumann entropy associated with a segment of future null infinity to be independent of the upper limit of the segment, we do *not* expect this entropy to be zero even when the lower limit is $-\infty$. This is because our current definition of the algebra, $\mathcal{B}_{-\infty,u_0}$ includes observables only at null infinity and therefore excludes observables that can detect massive particles. The state at null infinity is expected to be entangled with the quantum state of massive particles, and this is displayed by the finite intercept above.

We now discuss some aspects of the results above.

## 4.1 Discussion

**Failure of Page's argument**

The reader may be surprised at Figure 3, since it contradicts the common idea that the von Neumann entropy should obey a "Page curve" [137]. However, a little reflection will show that there is no real reason for surprise. *The Page curve is derived by assuming that the Hilbert space factorizes into the degrees of freedom inside the black hole and the degrees of freedom outside.* We already know that this assumption is not only wrong, it fails in the worst possible manner: holography implies that the information inside the black hole is just a copy of the information outside. There is no reason to expect that an expectation based on an incorrect assumption will correctly describe the von Neumann entropy of black hole radiation, and indeed it does not do so.

**Black hole evaporation in AdS**

The results above have an immediate analogue for anti-de Sitter space, which may also help the reader place them in a more familiar setting.

Consider a small black hole that evaporates in a spacetime with asymptotically AdS boundary conditions. The analogue of a cut at spatial infinity is now a cut of the timelike boundary. Corresponding to the algebra associated with the neighbourhood of a cut, we may now associate an algebra of asymptotic operators smeared over an $\epsilon$-extent in time.

The analysis of appendix A now tells us that if we measure all operators in this algebra, this completely specifies the state. Therefore, if we start with a pure state, and measure the von Neumann entropy of the asymptotic region, that von Neumann entropy always remains zero.

This result is obviously what is expected by holography. If we consider a small black hole that forms and evaporates with asymptotically AdS boundary conditions, the von Neumann entropy of the boundary does not obey the Page curve but instead remains fixed. This is just like Figure 3 except that in this case the constant value is also 0.[6]

**Cloning and strong subadditivity paradoxes**

Some versions of the information paradox are framed in terms of the "cloning" of information. For instance, if one considers a black hole formed from collapse, then it is possible to draw a nice slice that intersects the infalling matter and also a large fraction of the Hawking radiation. It is possible to argue that the late Hawking radiation must have information about the infalling matter. If one adopts a naive perspective on information-localization, where information on different parts of a spacelike slice is distinct, and then travels along causal trajectories, this leads to a paradox.

A closely related paradox, first described by Mathur [141] and later popularized by AMPS [142], is the strong subadditivity paradox. Here, the early Hawking radiation appears to be entangled both with the late Hawking radiation, and also the part of the black hole interior just behind the horizon. This appears to be in conflict with the monogamy of entanglement but it can again be viewed as the duplication of information. (We refer the reader to section 6.1 of [143] for a review of these paradoxes.)

However, from our perspective, not only is the resolution to the paradox clear, its physics is not even surprising. We have explained that the degrees of freedom localized on later cuts of null infinity are already contained in the degrees of freedom localized on earlier cuts. One can think of the degrees of freedom in the interior as corresponding to later cuts, and degrees

---

[6]It is worth nothing that this is also the result that follows by using the RT/HRT formula [138–140]. So, if one declares that the RT/HRT formulae are "semi-classical", then this gives an independent "semi-classical" derivation of the trivial von Neumann entropy curve.

of freedom in the early radiation as corresponding to earlier cuts. Therefore, it is clear that if one makes the mistake of thinking of these degrees of freedom as independent, this will lead to paradoxes involving cloning or the loss of the monogamy of entanglement. This resolution is, of course, not new and the same as the resolution suggested by the ideas of black hole complementarity [144–146] and ER=EPR [147].

## 4.2 The Page curve in AdS/CFT

We have argued above that the Page curve is not the correct curve for understanding the von Neumann entropy of black hole radiation in flat space or the radiation of a small black hole in AdS. We now explain how this is consistent with several recent papers that examine the Page curve in the context of black hole evaporation. *The key to obtaining a Page curve is to find a setting in which the Hilbert space factorizes*.

Conceptually, the simplest setting is to consider a plasma ball solution in AdS/CFT. The plasma ball is a black hole solution that is localized in some of the transverse directions, and therefore behaves like a "small black hole" in AdS/CFT. This solution was found explicitly in [148], and figure 4a shows a schematic representation. The plasma ball solution can be thought to be dual to a collection of a quark-gluon plasma that is localized in some region $D$ in the boundary gauge theory. This plasma slowly decays by emitting glueballs into the surrounding region $C$.

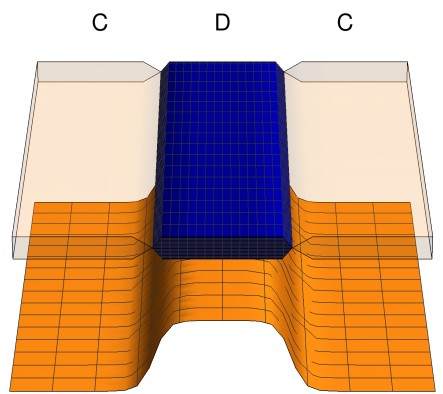

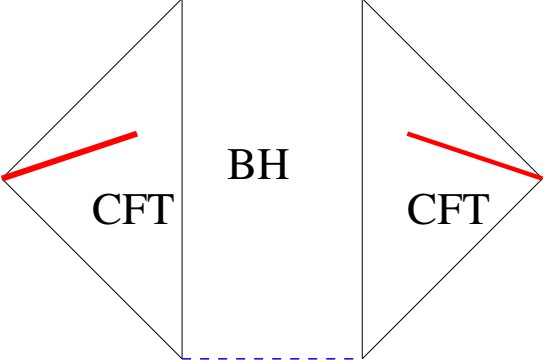

(a) A plasma ball solution is a black hole localized in the transverse dimensions (depicted by the localized yellow bump.) On the boundary, it corresponds to a localized lump of the deconfined phase (depicted by the blue region). Since the boundary Hilbert space factorizes, as the deconfined phase decays by emitting glueballs, the entropy of $C$ obeys a Page-like curve.

(b) A black hole in AdS coupled to a non-gravitational theory. Since the Hilbert space factorizes into the Hilbert space of the gravitational theory and the auxiliary CFT, it can be argued that the entanglement entropy of the red slices obeys a "semi-Page" curve that saturates at late times [10].

Figure 4

On the *boundary*, the Hilbert space now factorizes into the degrees of freedom in $C$ and its complement: $H_C \otimes H_D$. Therefore if one considers the entropy of the region $C$, we indeed expect it to obey the naive Page curve of Figure 2 as was explained in [145]. However, it

would be misleading to interpret this as the entropy of the black hole "radiation". The region $C$ carries information about a complicated combination of the exterior and the interior of the black hole in the bulk. But crucially, the region $C$ *excludes* some of the radiation that is near the region $D$. So this Page curve is best understood purely on the boundary: it is the standard Page curve of the boundary quantum field theory as energy in one region ($D$) spreads into the surrounding region ($C$).

The setups of [8–11] are all essentially of this type, and one setup is displayed in 4b. In all of them, we couple the CFT to an external system so that the Hilbert space again factorizes as $H = H_{CFT} \otimes H_{external}$. Once again the physics is best interpreted just on the boundary: we have some energy in a CFT, which we couple to an external reservoir. As the energy leaks from the CFT into the reservoir, it is indeed expected that the von Neumann entropy of the reservoir will obey the ordinary Page curve.

Once again, it is somewhat misleading to identify the reservoir as the radiation and the original CFT as the black hole. This is because the bulk dual of the reservoir necessarily excludes some of the radiation, which continues to be described by the original CFT. One way to bring out the significance of this point more vividly is to try and define what one includes in the "radiation" independently of the boundary description. For instance one reasonable definition of the radiation would be to consider the algebra of all operators that are localized outside the black hole. Such an algebra would always have a zero von Neumann entropy and it does *not* coincide with the algebra of operators that act purely on $H_{external}$.[7]

Leaving aside this question of interpretation, out results are not in contradiction with those of [8–11] since their setting is different.

**The Page curve in flat space?**

On the other hand, it was suggested in [149] that the setup of Figure 4b may have "phenomenological" applications in that the entropy of the radiation outside a near-extremal four-dimensional black hole in asymptotically flat space may obey a Page-like curve. However, in a real four-dimensional theory of gravity, our results suggest that the von Neumann entropy of Figure 4b should obey the trivial Page curve shown in Figure 3 and moreover that the constant value of this entropy should be 0. This is because, at each point of time, not only do the red slices of Figure 4b capture the region near $\mathcal{I}_-^+$, the bulk of these slices also captures the information present in massive degrees of freedom. Gravity may be weak on these slices but it would nevertheless be erroneous to ignore gravitational effects in the computation of the fine-grained von Neumann entropy. Indeed, it is precisely this erroneous assumption — the idea that when gravity is weak, we can ignore its novel storage of quantum information—that has led us into several paradoxes in the past including the cloning and strong subadditivity paradoxes.

So does the Page curve have any relevance for flat-space black holes? We would like to eschew discussions of "quantum computers" that "collect Hawking radiation", since such scenarios are extremely imprecise. But perhaps the Page curve is relevant for some subalgebra of $\mathcal{A}(\mathcal{I}^+)$. One possibility is to consider only the algebra of *news operators* discarding both the shear and the Bondi mass. This is not very natural, from a physical perspective, since all of these are just components of the metric. On the other hand, it has the advantage that the algebra of news operators on null infinity factorizes into a product of the algebras associated with its cuts. Indeed the news, at null infinity, can be decomposed through a spherical harmonic decomposition into an infinite set of free-fields. The formulas of [150] can then be used to

---

[7]If we think of gauge fixing operators by dressing them to the boundary, it is clear that there is no sense in which an operator can be localized purely inside the black hole since the dressing necessarily makes reference to the region outside. In contrast, there is a precise sense in which we can consider operators "outside" the black hole: these are operators whose location and dressing lies outside the horizon.

compute the von Neumann entropy of any segment of null infinity. But since the news operators commute with the soft charge, this restricted algebra of observables cannot extract the information present in the soft-part of the wavefunction. So these observations will always perceive the state to be a mixed state. It would be interesting to understand how much of the information is present in the soft-part of the wavefunction—an issue that was also discussed in [151].

Another interesting question is to examine how the von Neumann entropy of a segment of future null infinity varies with the *lower limit* of the segment. Once again, a quantitative analysis requires us to understand how much information is present in the soft-modes.

To the extent that the Page curve is an answer in search of a question, we do not rule out the possibility that an appropriate question can be found, in the context of flat space black holes, for which the answer is given by the Page curve. But, it seems to us, that rather than finding ways of justifying the Page curve, which is based on assumptions known to be incorrect, it may be more fruitful to focus on the striking and novel aspects of quantum information in quantum gravity.

# 5   Conclusion

**Summary of results**

In this paper, we have described how quantum information is stored holographically at null infinity. We showed, subject to reasonable physical assumptions, that all information about massless excitations could be obtained from observables in an infinitesimal neighbourhood near spatial infinity, even though it naively seems that such information requires observations over all of null infinity. With stronger assumptions, we showed that this information obeyed a nested structure: observables at a cut of future null infinity could always extract the same information as observables on a later cut. These results have direct implications for the information paradox: they imply that the von Neumann entropy of the state defined on a segment of null infinity $(-\infty, u)$ remains constant and so does not obey the Page curve. (As we noted, it may be possible to frame an alternate appropriate question for which the Page curve is an answer.)

**Comparison with other proposals for flat-space holography**

One way to understand the difference in focus in this paper with much of the extant literature on flat-space holography, is by analogy to AdS/CFT. AdS/CFT was first made precise by relating CFT correlators to asymptotic bulk correlators. This has been a remarkably fruitful program but, at leading nontrivial order, this map works even without making reference to gravity in the bulk. In fact, gravity enters only at a later stage because the CFT must have a stress-tensor that, in the bulk, must be dual to a graviton. On the other hand, as emphasized in the introduction, there is a striking fact about the storage of quantum information in asymptotically AdS spacetimes, which does crucially require gravity and stands somewhat apart from the program of matching bulk and boundary correlators. This is that bulk operators near the boundary, at any given time, already know about what is happening deep in the interior.

This is similar to the complementary relationship between the program initiated in this paper, of understanding how information is stored at null infinity, and other interesting questions such as the problem of rewriting flat-space S-matrix elements as correlators of a dual conformal field theory on the celestial sphere [15–17]. In fact, just as in AdS, the S-matrix/celestial CFT map appears not to use any features that are unique to gravity, whereas gravity is crucial for the results in this paper. One consequence of this is that the S-matrix/celestial CFT map uses information along the entire null boundary even though this information is, in principle,

accessible from just a thin slice in a theory of gravity.

On the other hand, since a complete description of the state is possible *both* at future and past null infinity, our results hold independently for these two null boundaries. So, while the results in this paper crucially rely on the constraints that come from gravity, they also do not make immediate contact with the dynamical aspects of bulk gravity. These dynamical aspects are what control the S-matrix, which tells us how in-states *map* to out-states. It would be nice to understand the constraints, if any, that our observations impose on the bulk dynamics.

Another natural question is whether flat-space holography can be understood by taking a limit of AdS/CFT. Here, we are not referring to the question of extracting perturbative S-matrix elements from AdS correlators, which has been studied extensively. A complete flat-space limit would involve keeping $g_s$ fixed, while taking the AdS radius to infinity in string units. It is clear that this takes us out of the 't Hooft limit; in the $\mathcal{N} = 4$ theory, for instance, this requires us to take the gauge theory coupling, $\lambda \to \infty$, while keeping $g_{\mathrm{YM}}$ finite. It is somewhat remarkable that the BMS algebra can indeed be recovered as a limit of a (non-relativistic) conformal algebra in this manner [13,14]. However, it seems to us that some simple geometric questions remain unclear. For instance, should the cylindrical boundary of AdS map to null infinity, or to the blowup of spatial infinity described in [152]? If the answer is the latter, it should be possible to generalize our results to argue that a theory on this blowup of spatial infinity contains not only all information about the bulk, but also captures all bulk dynamics.

Such a conjecture was, in fact, described by Marolf in a set of interesting papers [105–107]. While our work is similar in spirit, there are several important differences in our assumptions and results that we now explain. The analysis of [105–107] assumed that the Hamiltonian would remain a boundary term in the full theory of quantum gravity — this is similar to what we called assumption (2.1) — to arrive at a result analogous to our result (1). We have emphasized that the same result can be obtained through considerably weaker assumptions that rely only on the low energy structure of the theory and its vacua. Our argument for this result is entirely independent. Moreover, we argued that if one does make the stronger assumption that the commutators of canonical gravity can be carried over to the full theory, then one can obtain the novel result (2), which provides an interesting and intricate picture of the storage of information at null infinity.

To avoid confusion, we also clarify that our use of perturbation theory is different from its use in [106]. We mentioned that perturbation theory could be used to check our results by computing simple correlation functions involving the Bondi mass, as explained near equation (47). But, in [106], perturbation theory was used to relate operators on the past boundary of $\mathcal{I}^+$ to the future boundary of $\mathcal{I}^-$. This latter proposal is much more ambitious since it seems to use perturbation theory to relate the future boundary of the blowup of spatial infinity to its past boundary, which requires evolution for an infinite amount of time.

### A broader program

The results (1) and (2) hint at the following general picture in quantum gravity. In a local quantum field theory, we are free to specify the quantum wavefunction of a state separately on different parts of a spatial slice. For instance, referring to Figure 5, in a nongravitational theory, we are free to put some feature in the middle of the spatial slice without affecting the wavefunction outside a bounded region. However, in gravity, our results suggest that the constraints are so strong that if we specify the wavefunction everywhere outside a bounded region, we also specify it inside that region!

The picture above seems rather robust for asymptotically AdS spacetimes. But while we have made some progress in this paper, we have not completely established such a picture for asymptotically flat spacetimes. One key missing ingredient is the treatment of massive particles. While classical data for massive particles cannot be specified at null infinity, we

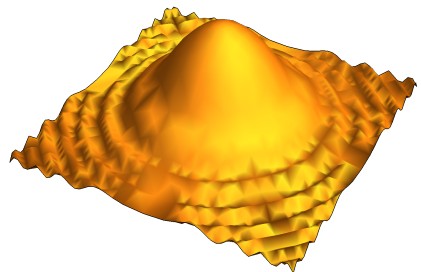

Figure 5: In ordinary local quantum field theories we can specify some feature in the wavefunction inside a bounded region (depicted by the bump) without affecting the wavefunction outside. But, in gravity, not only is one forced to modify the wavefunction outside the region (shown by the wiggles), the constraints are so strong that the detailed wavefunction outside completely fixes it inside.

suspect that it should be possible to include massive particles in our framework if, instead of looking at null infinity, we consider a "thickened boundary" that also includes an infinitesimal portion of the bulk.

**Future directions**

Apart from the inclusion of massive particles, a clear future direction is to extend our story to other spacetime dimensions. There is some debate in the literature on the vacuum structure of gravity in other spacetime dimensions. However, regardless of the answer to this question, it appears likely that the vacuum— or degenerate vacua, as the case may be— should be identifiable by charges supported near spatial infinity. If so, the program outlined in this paper should carry through to other spacetime dimensions.

The results that we present here are the flat-space analogues to the statement that a cut of the AdS boundary contains all information about the bulk. However, there are more fine-grained statements that can be made in AdS/CFT: the entanglement wedge conjecture is that a subregion on the boundary contains information about the entanglement wedge in the bulk. How can this conjecture be understood from a canonical perspective? And what is its flat space analogue? These appear to be interesting questions for future work.

## Acknowledgments

We are grateful to Bidisha Chakrabarty for collaboration in the early stages of this work. We are also grateful to Abhay Ashtekar, Miguel Campiglia, Chandramouli Chowdhury, Victor Godet, Nima Lashkari, R. Loganayagam, Shiraz Minwalla, Pranav Pandit, Kyriakos Papadodimas, Kartik Prabhu, Biswajit Sahoo, Ashoke Sen, Madhavan Varadarajan and Edward Witten for helpful discussions. We would also like to thank all the participants of the Utrecht Triangle Meeting on Holography, the National Strings Meeting (Bhopal) and the Chennai Symposium on Gravitation and Cosmology, where preliminary versions of these results were presented, for their helpful feedback and discussions. S.R. is partially supported by a Swarnajayanti fellowship, DST/SJF/PSA-02/2016-17, of the Department of Science and Technology.

# Appendix

## A  A canonical perspective on AdS holography

In this appendix, we provide a lightning review of a canonical perspective on holography with asymptotically anti-de Sitter boundary conditions. This appendix largely follows the treatment of [5] but we sharpen some of the assumptions in that analysis, and elaborate on some intermediate steps. We have written this Appendix to be self-contained and so there is some overlap with the discussion in the main text.

We would like to establish the following statement

**Result 5.** *If, in asymptotically anti-de Sitter space, two states $|\Psi_1\rangle$ and $|\Psi_2\rangle$ are distinct then, in a theory of quantum gravity, they can be distinguished purely through asymptotic operators in an infinitesimal time band.*

This shows that all quantum information about the state lies within asymptotic operators. We will make the statement precise below, and prove it, subject to some plausible assumptions. As in the main text, this discussion should be understood from the following perspective. Even if we do not know the details of the full UV-complete theory of gravity, using low energy observations and some reasonable extrapolations (which we detail precisely below), it is possible to conclude that the full theory will be holographic.

**Setup and boundary conditions**

The metric of global AdS in $d+1$ spacetime dimensions is given by

$$g_{\mu\nu}^{\text{AdS}}dx^\mu dx^\nu = -(r^2+1)dt^2 + \frac{dr^2}{r^2+1} + r^2 d\Omega_{d-1}^2. \tag{55}$$

We are interested in spacetimes that may differ from this spacetime at finite $r$ but tend to global AdS asymptotically. More precisely, we consider metrics of the form

$$g_{\mu\nu} = g_{\mu\nu}^{\text{AdS}} + h_{\mu\nu}. \tag{56}$$

As $r \to \infty$, we choose the Fefferman Graham gauge $h_{r\mu} = 0$, and on the remaining components of the metric, we impose the boundary conditions

$$h_{ij} \xrightarrow[r\to\infty]{} \frac{1}{r^{d-2}}. \tag{57}$$

We may have other propagating degrees of freedom in the theory, and their boundary falloffs are set as is standard in AdS/CFT. For instance, if we consider a scalar field of mass $m$, then we demand that the fluctuations of this field die off asymptotically as

$$\phi \xrightarrow[r\to\infty]{} \frac{1}{r^\Delta}, \tag{58}$$

where $\Delta = \frac{d}{2} + \left(\frac{d^2}{4} + m^2\right)^{\frac{1}{2}}$. This corresponds to what is usually called the "normalizable" mode in AdS/CFT.

The boundary of the spacetime considered above has topology $S^{d-1} \times R$, and we use the coordinates $(\Omega, t)$ to specify a point on the boundary.

**Physical observables and boundary algebra**

Good physical observables in a theory of quantum gravity are those that are invariant under *small diffeomorphisms* i.e. those diffeomorphisms that die off near the boundary. In particular, the *boundary values* of propagating fields give a set of well-defined physical observables. The asymptotic fluctuations of the metric yield the observables

$$t_{ij}(\Omega, t) = \lim_{r \to \infty} r^{d-2} h_{ij}(r, \Omega, t). \tag{59}$$

The asymptotic fluctuations of a scalar field also yield observables through

$$O(\Omega, t) = \lim_{r \to \infty} r^\Delta \phi(r, \Omega, t). \tag{60}$$

The observables above are labeled by a time and a point on the $S^{d-1}$.

Now consider the set of all possible functions of such observables in a small time band $(0, \epsilon)$ but with arbitrary values of $\Omega$. We will call this algebra $\mathcal{A}_{\text{bdry}}$. Some of its lowest order terms are

$$\begin{aligned}
\mathcal{A}_{\text{bdry}} = \{ & t_{i_1 j_1}(t_1, \Omega_1), O(t_1, \Omega_1), t_{i_1 j_1}(t_1, \Omega_1) t_{i_2 j_2}(t_2, \Omega_2), \\
& t_{i_1 j_1}(t_1, \Omega_1) O(t_2, \Omega_2), O(t_1, \Omega_1) O(t_2, \Omega_2) \ldots \},
\end{aligned} \tag{61}$$

where all the $t_i \in (0, \epsilon)$ and $\epsilon$ can be any finite number.

**The Hilbert Space**

We now describe the Hilbert space of the theory. An analysis of low-energy fluctuations about the global AdS metric tells us that this Hilbert space has a *unique vacuum*, which is separated from the nearest excited state by a gap.

**Assumption 5.1.** *We assume that in the full theory of quantum gravity, this low energy structure is preserved. In particular, we assume that the full theory has a unique vacuum, which we denote by $|0\rangle$.*

We now consider the space of states obtained by exciting this vacuum with all possible asymptotic operators at *arbitrary values* of time and with arbitrary coordinates on the $S^{d-1}$. This leads to the Hilbert space

$$\begin{aligned}
\mathcal{H} = \{ & t_{i_1 j_1}(t_1, \Omega_1)|0\rangle, O(t_1, \Omega_1)|0\rangle, t_{i_1 j_1}(t_1, \Omega_1) t_{i_2 j_2}(t_2, \Omega_2)|0\rangle, \\
& t_{i_1 j_1}(t_1, \Omega_1) O(t_2, \Omega_2)|0\rangle, O(t_1, \Omega_1) O(t_2, \Omega_2)|0\rangle \ldots \},
\end{aligned} \tag{62}$$

where $t_i \in (-\infty, \infty)$ and $\Omega_i$ are points on the $S^{d-1}$.

The set of operators that appear in (62) may appear to be very similar to those that appear in (61). However, there is a *crucial difference*: in the definition of the Hilbert space, (62), we have *not* restricted $t_i$ to a small time band (as we did while defining the boundary algebra in (61)) but instead have allowed $t_i$ to range over all possible real values.

Physically this Hilbert space corresponds to states that can be generated through the following process: we start with the vacuum in the far past, and then excite the vacuum by means of asymptotic operators. This space of states is very large and, in particular, includes all black holes that can be formed from collapse.

**Closure under time evolution**

The key mathematical reason why we expect to be able to completely formulate the theory within the Hilbert space (62), is that it is *manifestly* closed under time-evolution. Consider evolving the set of states in (62) with the Hamiltonian of the full theory. Even though this

Hamiltonian may be very complicated, since $|0\rangle$ is the vacuum of this Hamiltonian, this time-evolution only shifts the coordinates $t_i$. Since we have allowed all possible values of $t_i$ in the definition (62), we see that time-evolution *cannot* take us out of the space above. So, *considerations of unitarity, by themselves, cannot force us to enlarge the Hilbert space defined in* (62).

This is a key difference from flat space. In flat space, one must be very careful in restricting the Hilbert space either on future or past null infinity since unitarity requires the two spaces to be mapped to each other under evolution through the bulk. But, in AdS, which has a single boundary, it appears very reasonable to make the following assumption.

**Assumption 5.2.** *We assume that the theory can be completely formulated within the Hilbert space $\mathcal{H}$.*

**The Hamiltonian**

Within effective field theory, the Hamiltonian of the theory is itself an asymptotic operator. A standard analysis tells us that the effective field theory Hamiltonian is given by

$$H = \frac{d}{16\pi G} \int d^{d-1}\Omega\, t_{tt},\qquad(63)$$

where $t_{tt}$ is the boundary value of the metric fluctuation defined above [153].

The fact that the semiclassical Hamiltonian is given by a boundary term is expected. In the standard canonical analysis of gravity, we consider wavefunctions that are invariant under small diffeomorphisms. This is reflected in the Hamiltonian constraint. This directly implies that, on any valid wavefunction (i.e. one that satisfies the Hamiltonian constraints), the Hamiltonian reduces to a boundary term.[8]

We now make an important assumption.

**Assumption 5.3.** *We assume that the asymptotic operator* (63)*, remains a positive operator in the full theory of quantum gravity, and moreover that its vacuum coincides with the exact vacuum, $|0\rangle$, of the full theory.*

Note that we are *not* assuming that formula above, (63), is an exact formula for the Hamiltonian in the full quantum gravity theory. Rather we are only assuming that the boundary value of the metric gives us the correct Hamiltonian within the space of low-energy states and, in particular, that it identifies the correct vacuum for us. This is only an assumption about *low energy physics*. Moreover, if even this assumption holds only approximately and not exactly then our statements below will continue to be valid at the same approximate level.

As we explained in the main text, if an operator belongs to the algebra then so do all of its spectral projectors. In particular the projector on the vacuum of $H$ belongs to the algebra, $\mathcal{A}_{\text{bdry}}$. Since this projector is the same as the projector onto the vacuum of the full Hamiltonian, our assumption above implies that

$$P_0 = |0\rangle\langle 0| \in \mathcal{A}_{\text{bdry}}.\qquad(64)$$

**Squeezing the generators of the Hilbert space**

The Hilbert space above was generated by acting with asymptotic generators at arbitrary time values on the vacuum, and we argued that this was sufficient to formulate the theory since it was manifestly closed under Hamiltonian evolution.

---

[8]For the reader who is familiar with AdS/CFT, we would like to point out that the formula (63) is just a manifestation of the "extrapolate" dictionary. The boundary value of the metric fluctuation, $t_{ij}$ is dual to the CFT stress-tensor and integrating the stress-tensor gives us the Hamiltonian. However, we emphasize that we are not assuming this dictionary anywhere in this Appendix and our reasoning just follows canonical principles.

We now argue that same space can be generated by the action of the boundary algebra defined in (61) on the vacuum.

$$\mathcal{H} = \mathcal{A}_{\text{bdry}}|0\rangle. \tag{65}$$

Although this may seem surprising at first sight since the boundary algebra only contains operators in the infinitesimal time band $(0, \epsilon)$, this statement is not difficult to prove.

We will prove the claim above through contradiction. Imagine that there exists some state, $|\Psi\rangle \in \mathcal{H}$, that is orthogonal to all states generated by the action of the boundary algebra on the vacuum. This means that

$$C(t_1 \ldots t_n) = \langle \Psi | O(t_1, \Omega_1) O(t_2, \Omega_2) \ldots O(t_n, \Omega_n) | 0 \rangle = 0, \tag{66}$$

for any $t_i \in (0, \epsilon)$. Now by inserting a complete set of energy eigenstates in between the operators, and by *assuming the positivity of the full Hamiltonian,* we find that

$$C(t_1 \ldots t_n) = \sum_{E_i} e^{i \sum_j E_j z_j} \langle \Psi | E_1 \rangle \langle E_1 | O(0, \Omega_1) | E_2 \rangle \ldots \langle E_n | O(0, \Omega_n) | 0 \rangle, \tag{67}$$

where the variables $z_j$ are given by

$$z_1 = t_1; \quad z_2 = t_2 - t_1; \quad \ldots \quad z_n = t_n - t_{n-1}. \tag{68}$$

It is clear that the function $C$ is analytic when all the $z_j$ are extended in the *upper half plane.*

But now if $C$ vanishes when all the $t_i \in (0, \epsilon)$, by the edge-of-the-wedge theorem, it must vanish for *all real $t_i$.* But this is impossible, since we assumed that $|\Psi\rangle$ was an element of $\mathcal{H}$, and so it was created by the action of asymptotic operators at some times. This implies that $|\Psi\rangle$ cannot exist.

Therefore all the states in the Hilbert space can be created by the action of asymptotic operators in an infinitesimal time interval.

**Completing the proof**

We now move to the final step in our argument. We want to prove that if we have two distinct states, $|\Psi_1\rangle \neq |\Psi_2\rangle$, then we can find an asymptotic operator within the infinitesimal time band $(0, \epsilon)$ that can distinguish these two states.[9]

Since the states are distinct, there must exist some operator $Q$ that can distinguish between them, i.e. $\langle \Psi_1 | Q | \Psi_1 \rangle \neq \langle \Psi_2 | Q | \Psi_2 \rangle$.

Now, we can expand $Q$ in some basis of states, which we denote by $|n\rangle$, so that

$$Q = \sum_{nm} c_{nm} |n\rangle \langle m|. \tag{69}$$

By the argument above, each element in the basis can be generated by the action of some element of $\mathcal{A}_{\text{bdry}}$ on the vacuum. So

$$|n\rangle = X_n |0\rangle; \quad |m\rangle = X_m |0\rangle, \qquad X_n, X_m \in \mathcal{A}_{\text{bdry}}. \tag{70}$$

Therefore

$$Q = \sum_{nm} c_{nm} X_n |0\rangle \langle 0| X_m^\dagger = \sum_{nm} c_{nm} X_n P_0 X_m^\dagger. \tag{71}$$

---

[9]We emphasize that being able to distinguish all states through an operator in $\mathcal{A}_{\text{bdry}}$ is *much stronger* than the statement that all states can be generated by the action of $\mathcal{A}_{\text{bdry}}$ on the vacuum. For instance, if we consider ordinary local quantum field theory in flat space, then all states can be generated by the action of the algebra belonging to any open set. But of course, it is *not true* that in local quantum field theories, all states can be distinguished through observables from the algebra from any open set.

But since $X_n, P_0, X_m$ all belong to $\mathcal{A}_{\text{bdry}}$ and since $\mathcal{A}_{\text{bdry}}$ is an algebra, which must be closed under products and linear combinations, the entire sum on the right hand side belongs to $\mathcal{A}_{\text{bdry}}$. So $Q \in \mathcal{A}_{\text{bdry}}$. Therefore any distinct states can be distinguished by the action of an element of $\mathcal{A}_{\text{bdry}}$. ∎.

Most aspects of the discussion that we presented in flat space also applies here. The argument above does not work in a non-gravitational theory, because it is *only* in a theory of gravity that the projector on the vacuum can be written as an asymptotic observable on a single time slice. So, only gravitational theories are holographic whereas nongravitational theories, including gauge theories, are not. Moreover, the argument above does not work classically. Finally, as we will explain in [134] this argument can be verified in perturbation theory. This perturbative verification is impossible unless we take $\epsilon > 0$. This is the reason that we consider asymptotic operators in a finite time-band rather than asymptotic operators at just an instant of time.

# B    Asymptotic charges as observables

In this section, we will address the argument of [108] and explain how the ADM mass and other asymptotic charges should be viewed as observables at null infinity.

The argument of [108] is quite simple, and can be understood by considering even a free scalar theory. Usually, we define the vacuum as an eigenstate of the Hamiltonian which (after a possible shift in the zero-point point energy) annihilates it.

$$H|0\rangle = 0. \tag{72}$$

In a local theory, the operator $H$ above is an integral of a Hamiltonian density, $\mathcal{H}(\vec{x}, t)$ over an entire Cauchy slice. However, now consider integrating the Hamiltonian density only inside some ball of large radius, $B(R)$,

$$H(R, t) = \int_{B(R)} \mathcal{H}(\vec{x}, t) d^3 \vec{x}. \tag{73}$$

The point made in [108] is that this truncated operator has large fluctuations: in fact, $\langle 0|H(R)^2|0\rangle$ grows with $R$ and so it appears that a naive $R \to \infty$ limit will of $H(R)$ will not yield the correct Hamiltonian, $H$.

Now, if one couples the scalar to gravity, the gravitational constraints relate some components of the metric on a sphere of radius $R$ to the energy contained inside the sphere. Therefore [108] argued that these metric components would also have large fluctuations that could only be tamed by smearing $H(R)$ over a large time.

In this Appendix, we show that the large quantum fluctuations can also be avoided by smearing $H(R, t)$ *radially* before taking the $R \to \infty$ limit. So to obtain asymptotic charges at null infinity, we first smear the bulk metric radially and then take its large-radius limit. We discuss the Hamiltonian in this section, but the same prescription can be used to define any supertranslation charge by integrating the radially smeared metric with an appropriate spherical harmonic.

Another way to view this is to recognize that interval $u \in (-\infty, -\frac{1}{\epsilon})$ actually contains an infinite amount of retarded time. Since the retarded time is $u = t - r$, by smearing over a large radial extent we are taking advantage of this to obtain a well-defined notion of the ADM mass and other asymptotic charges at $\mathcal{I}_-^+$. In forthcoming work, we will explore the utility of a similar smearing prescription for defining the exponentiated Bondi mass operator in equation (44).

The ADM mass defined in this manner is related by the constraints to the following operator $H$.

$$H = \lim_{R \to \infty} H_{\mathrm{sm}}(R) = \lim_{R \to \infty} \int dt' dR' \mathfrak{g}(t') \mathfrak{F}_R(R') \int_{B(R')} \mathcal{H}(\vec{x}, t) d^3\vec{x}. \tag{74}$$

The smearing in time is controlled by $\mathfrak{g}(t)$ and has support over a small user-defined length scale, $\eta$ around $t = 0$. The smearing function in the radial direction, $\mathfrak{F}_R$, varies the radius of the ball in the range $R \pm R^{1-\delta} \eta^{\delta}$, where $\delta$ can be chosen to be any number satisfying $0 < \delta < \frac{1}{3}$. We show that the fluctuations of $H_{\mathrm{sm}}(R)$ in a massless scalar field theory are then suppressed as

$$\langle 0 | H_{\mathrm{sm}}(R)^2 | 0 \rangle = \frac{1}{120\pi} \frac{1}{R\eta} \left( \frac{R}{\eta} \right)^{3\delta}, \tag{75}$$

up to terms that fall off even faster with $R$. So this has a good limit as $R \to \infty$. We derive this result below.

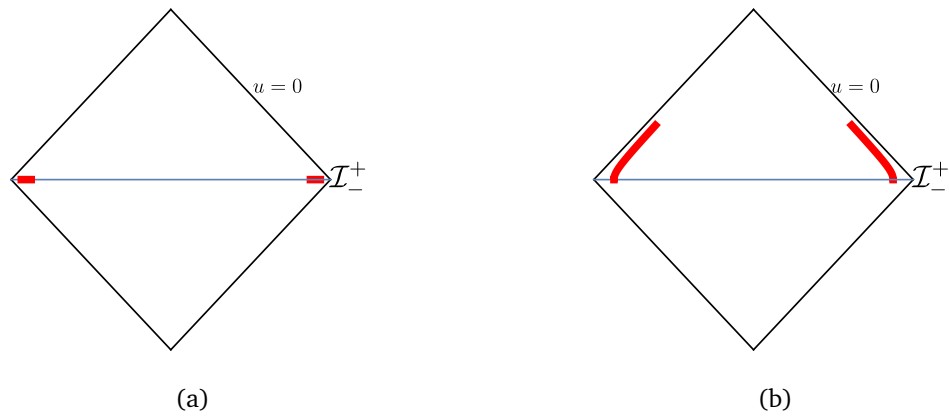

(a)                                      (b)

Figure 6: Our protocol (left) reduces quantum fluctuations by averaging over the radial direction. This allows us to associate asymptotic charges at $\mathcal{I}^+_-$ by smearing the metric over the thick red region. In particular, we avoid having to smear the metric up to a finite value of $u$ on $\mathcal{I}^+$ (right), which would have made the asymptotic charges ill-defined at $\mathcal{I}^+_-$.

## B.1 Fluctuations of the smeared Hamiltonian

Consider the free massless scalar field in $3 + 1-$dimensional spacetime.

$$\phi(x) = \int \frac{d^3k}{(2\pi)^{3/2}} \frac{1}{\sqrt{2\omega_{\vec{k}}}} \left[ a_{\vec{k}} e^{ik \cdot x} + a_{\vec{k}}^\dagger e^{-ik \cdot x} \right]. \tag{76}$$

The field satisfies canonical commutation relations provided

$$[a_{\vec{k}}, a_{\vec{k}'}^\dagger] = \delta^3(\vec{k} - \vec{k}'). \tag{77}$$

The Hamiltonian density is

$$\mathcal{H}(x, t) = \frac{1}{2} : \left[ (\partial_t \phi(x))^2 + (\vec{\nabla}\phi(x))^2 \right] :. \tag{78}$$

As usual, we normal order the Hamiltonian so that the divergent contribution of the zero-point energies is removed. Using the mode expansion we find

$$H(R,t) = -\frac{1}{2}\int_{|\vec{x}|\leq R} d^3x \frac{d^3k\,d^3p}{(2\pi)^3}\frac{\left(\omega_{\vec{k}}\,\omega_{\vec{p}} + \vec{k}\cdot\vec{p}\right)}{2\sqrt{\omega_{\vec{k}}\,\omega_{\vec{p}}}}$$
$$\left[a_{\vec{k}}\,a_{\vec{p}}\,e^{i(k+p)\cdot x} + a_{\vec{k}}^{\dagger}\,a_{\vec{p}}^{\dagger}\,e^{-i(k+p)\cdot x} - a_{\vec{k}}^{\dagger}\,a_{\vec{p}}\,e^{-i(k-p)\cdot x} - a_{\vec{p}}^{\dagger}\,a_{\vec{k}}\,e^{i(k-p)\cdot x}\right]. \tag{79}$$

Usually, we integrate over all $\vec{x}$ to define the Hamiltonian and then we just drop the terms with two creation and two annihilation operators. However, if we integrate over a finite region, these terms remain and the discussion here has to do with their effect.

$$H(R,t) = \int \frac{d^3k\,d^3p}{(2\pi)^2}\frac{\left(\omega_{\vec{k}}\omega_{\vec{p}} + \vec{k}\cdot\vec{p}\right)}{2\sqrt{\omega_{\vec{k}}\omega_{\vec{p}}}}\left(\mathcal{D}(k-p,R,t)a_{\vec{p}}^{\dagger}a_{\vec{k}} - \mathcal{D}(k+p,R,t)a_{\vec{k}}a_{\vec{p}} + \text{h.c.}\right), \tag{80}$$

with

$$\mathcal{D}(q,R,t) = \frac{\sin(|\vec{q}|R) - |\vec{q}|R\cos(|\vec{q}|R)}{|\vec{q}|^3}e^{-iq^0 t}. \tag{81}$$

In the large $R$ limit, the function $\mathcal{D}(q,R,t)$ is sharply peaked around $|\vec{q}| \to 0$.

As discussed earlier, we are interested in the smeared operator (74). We make the following choices for the smearing functions.

$$\mathfrak{F}_R(R') = \frac{1}{\lambda\sqrt{\pi}}e^{-\frac{(R'-R)^2}{\lambda^2}}; \qquad \mathfrak{g}(t) = \frac{1}{\eta\sqrt{\pi}}e^{-\frac{t^2}{\eta^2}}, \tag{82}$$

where

$$\lambda = R^{1-\delta}\eta^{\delta}, \tag{83}$$

and $\eta$ is chosen to be a small length scale that does not scale with $R$.

Now we proceed to the computation of fluctuations of smeared Hamiltonian.

$$\langle 0|H_{\text{sm}}^2(R)|0\rangle = \int dt_1\,dt_2\,dR_1\,dR_2\,\mathfrak{F}_R(R_1)\,\mathfrak{F}_R(R_2)\,\mathfrak{g}(t_1)\,\mathfrak{g}(t_2)\,\langle\Omega|H(R_1,t_1)H(R_2,t_2)|\Omega\rangle. \tag{84}$$

Expanding out both factors of $H$ in creation and annihilation operators, we see that the only term that contributes towards vacuum fluctuations is the one which picks up creation operators from the second factor and annihilation operators from the first. This leads to correlators of the form $\langle 0|a_{\vec{k}}a_{\vec{p}}a_{\vec{k}'}^{\dagger}a_{\vec{p}'}^{\dagger}|0\rangle = \delta(\vec{k}-\vec{k}')\delta(\vec{p}-\vec{p}') + \delta(\vec{k}-\vec{p}')\delta(\vec{p}-\vec{k}')$. We then find

$$\langle\Omega|H_{\text{sm}}^2(R)|\Omega\rangle = \int \frac{d^3k\,d^3p}{(2\pi)^4}\frac{\left(\omega_{\vec{k}}\omega_{\vec{p}} + \vec{k}\cdot\vec{p}\right)^2}{2\omega_{\vec{k}}\omega_{\vec{p}}}\mathcal{D}(k+p,R_1,t_1)\mathcal{D}^*(k+p,R_2,t_2)$$
$$\times \mathfrak{F}_R(R_1)\mathfrak{F}_R(R_2)\mathfrak{g}(t_1)\mathfrak{g}(t_2)dR_1dR_2dt_1dt_2. \tag{85}$$

Now we will perform all the smearing integrals to get the following.

$$\langle\Omega|H_{\text{sm}}^2(R)|\Omega\rangle = \int \frac{d^3k\,d^3p}{(2\pi)^4}\frac{\left(\omega_{\vec{k}}\omega_{\vec{p}} + \vec{k}\cdot\vec{p}\right)^2}{2\omega_{\vec{k}}\omega_{\vec{p}}}e^{-\frac{1}{2}\eta^2(\omega_{\vec{k}}+\omega_{\vec{p}})^2}\frac{1}{4}e^{-\frac{1}{2}\lambda^2(|\vec{k}+\vec{p}|)^2}\frac{1}{|\vec{k}+\vec{p}|^6}$$
$$\times \left[(2 + \lambda^2(|\vec{k}+\vec{p}|)^2)\sin(|\vec{k}+\vec{p}|R) - 2|\vec{k}+\vec{p}|R\cos(|\vec{k}+\vec{p}|R)\right]^2. \tag{86}$$

To simplify the above integral, we change variables to $\vec{q} = \vec{k} + \vec{p}$ and $\vec{r} = \vec{k} - \vec{p}$. Then we see that the integral above only receives contributions from the range where $|\vec{q}|\eta \ll 1$. This

allows us to series expand all terms in a series in $|\vec{q}|$ (except for those that involve $|\vec{q}|R$) and do the integrals explicitly leading to

$$\langle 0|H_{\mathrm{sm}}^2(R)|0\rangle = \frac{R^2}{120\pi\eta\lambda^3}\left(1 + \mathrm{O}\left(\frac{\lambda}{R}\right)\right) = \frac{1}{120\pi}\frac{1}{R\eta}\left(\frac{R}{\eta}\right)^{3\delta}\left(1 + \mathrm{O}\left(\frac{\lambda}{R}\right)\right), \qquad (87)$$

as advertised.

While we also required a small smearing over time, this only provided us with a UV-momentum cutoff. However, our radial smearing was over a *parametrically* larger region. This provided us with another momentum cut-off that was parametrically smaller than the cutoff provided by the time-smearing. This is the crucial difference with [108]. In fact, we can recover the results of [108] just by taking $\lambda = \mathrm{O}(\eta)$ in our calculation. Then the fluctuations of our smeared Hamiltonian again start to diverge with $R$ as is evident above.

The discussion here has had to do with the *definition* of asymptotic observables as limits of bulk observables and not with the "difficulty" of measuring them practically. To summarize, we do not see any, in-principle, obstacle to taking this limit and obtaining asymptotic charges as observables in a quantum theory, provided the limit is taken carefully.

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
