# Peer review of "The Holographic Nature of Null Infinity"

_SciPost Physics, doi:SciPost Phys. 10, 041 (2021)_

## Round 2 · Referee Report · Daniel Harlow (Referee 1) · 2021-1-27

Report

This is an interesting paper which shows that under various fairly plausible assumptions about the nature of non-perturbative quantum gravity some kind of holography is inevitable. The logic has much in common with earlier work by Marolf, see in particular 0808.2842, but the authors explain things more clearly and also go further. On the other hand I think in some places they go too far: they do not spell out all of their implicit assumptions, and their criticism of the Page curve is I believe somewhat off base. I will now explain these comments in more detail.

The authors have very kindly presented the AdS versions of their arguments in appendix A, and I will take advantage of this to highlight my concerns in this relatively well-understood arena (similar comments apply to the asymptotically-flat case). The key assumptions which I believe are necessary to make the argument the authors which to make are the following:

1) The boundary limits of low-energy bulk fields (as in equations A.5,A.6) make sense as well-defined operators on the full Hilbert space of non-perturbative quantum gravity. 2) The exact Hamiltonian is bounded from below and has a unique ground state. 3) The algebra generated by these boundary limits of low-energy fields acts cyclically on the ground state. 4) The exact Hamiltonian is equivalent to the ADM Hamiltonian, which makes sense on the full Hilbert space due to the first assumption.

The authors discuss assumptions 2) and 3) in some detail, but I think they downplay the importance of assumption 1) (which they do not even call attention to), and they suggest that assumption 4) can be usefully weakened while I do not believe it really can.

To see the importance and nontriviality of assumption 1), I emphasize that the Hilbert space defined by equation A.8 is NOT well-defined within low-energy effective field theory. The reason is that in order to compute the inner products between these states we need to know the vacuum correlators of the low-energy fields at large time separation, in which case the perturbative expansion in the gravitational coupling breaks down and non-perturbative control is necessary. Since everything else proceeds from this construction of the Hilbert space, all of the nontrivial results will depend sensitively on the properties of the UV completion (see the next paragraph for more on why the dependence is sensitive). In particular any attempt to distinguish generic states by operators in a time strip at future infinity will need to make use of the full non-perturbative structure of the theory. It is absolutely NOT the case that holography is a consequence of the gravitational constraints alone, and assumptions 1)-4) make it clear what else is necessary. Indeed in low dimensions there are explicit examples of renormalizable AdS quantum gravity which are NOT holographic (such as the canonical quantization of JT coupled to conformal matter or pure 2+1 gravity), and I leave it as an exercise for the reader to identify which of assumptions 1-4 fail in those examples.

As for assumption 4), the authors suggest that it can be weakened to only require that the exact and ADM Hamiltonians have the same ground state, and possibly even further to allow these ground states to only be approximately equivalent. The former assertion is technically correct, but the general opinion is that outside of integrable models it is impossible for two distinct local Hamiltonians to have the same ground state (see for example 1712.01850 by Ranard and Qi). The latter however is quite unjustified: the argument of appendix A makes use of very delicate properties of von Neumann algebras and holomorphic functions, and there is absolutely no reason to assume it would go through in an approximate situation. In particular the application of the edge of the wedge theorem makes crucial use of the fact that a holomorphic function which is zero on a line interval is zero everywhere, while it is certainly not the case that a holomorphic function which is small on a line interval is small everywhere! Similarly approximate statements about operators are notoroiusly finicky, especially in an infinite-dimensional setting. For my money the authors would do better to just adopt assumption 4), the results are already interesting enough without trying to weaken it.

Turning now to the criticism of the usual discussion of the Page curve, I believe the authors are over-interpreting their results. In AdS/CFT it is certainly true that an outside observer with full access to the CFT degrees of freedom at a fixed time can determine all properties of the bulk state. But this does not mean that approximate locality in the bulk is violated for simple experiments in appropriately semiclassical states. The kind of operators promised by result 5 on page 30 would necessarily involve products of very large numbers of the bulk fields near the boundary, and whatever process was used to measure them would surely involve the creation of very large black holes which would destroy the semiclassical picture of the bulk. As explained in 1411.7041, the local structure in the bulk is only respected if one restricts to operations which preserve a sufficiently small "code subspace". In particular if we consider the Hawking cloud produced by the evaporation of a small black hole in AdS, this can be viewed as living in a Fock space of (dressed) quanta which can be thought of as part of a code subspace which is still small enough that there is no danger of forming a large black hole. This locality is only approximate, but the approximation becomes quite good at large $N$ and in particular it can be much MORE accurate than $e^{-S}$ where $S$ is the entropy of the small black hole.

In other words, the authors are not being fair in demanding that the Page curve be formulated as an exact notion in terms of the exact boundary algebra. It is true that sometimes this can be achieved, as in the recent calculations where AdS is coupled to a non-gravitational system, but in more realistic situations such as a small black hole evaporating within AdS or an asymptotically-flat black hole the Page curve has to be understood as a statement about approximate "bulk entropy" instead of a statement about exact "boundary entropy". The most concrete way to think about this, which the authors unfortunately dismiss on page 26, is to think about what would happen if a bulk observer were to collect all of the Hawking radiation and feed it into a bulk quantum computer. The authors may find this question "extremely imprecise", but in fact this is the only way one could ever test the unitarity of black hole evaporation in practice! And they owe us an answer: do they think that the state of the quantum computer would have a Page curve like the one in figure 2, or do they think it would have one like the Page curve in figure 3? Myself I'll go with figure 2.

These comments aside, I think it is fine for the paper to be published as is. These are research-level disagreements, not trivial errors, and the authors are welcome to incorporate them if they wish but also welcome to leave the paper unmodified.

  • validity: -
  • significance: -
  • originality: -
  • clarity: -
  • formatting: -
  • grammar: -

Author:  Suvrat Raju  on 2021-02-05  [id 1204]

(in reply to Report 1 by Daniel Harlow on 2021-01-27)

We are grateful to Prof. Harlow for his insightful comments.

Before we respond in detail, we wold like to note that Prof. Harlow's report focuses on the results of Appendix A, which deals with the generalization of our results to asymptotically AdS spacetimes. But our main objective was to elucidate the precise sense in which flat space is holographic. Our main results are presented in Result 1 and Result 2. Result 1 states that information available on future null infinity is also available near its past boundary whereas Result 2 states (under stronger assumptions than result 1) that information available on a cut of future null infinity is available on any cut to its past.

Subtleties in the Hilbert space.

Prof. Harlow notes that there might be subtleties in constructing the Hilbert space by acting on the vacuum with boundary operators. As Referee 3 also notes, there are some differences between flat space and AdS in this regard.

In flat space, the Hilbert space at null infinity is simply a Fock space. The S-matrix, which makes sense even in string theory, is the overlap of states in the Fock space at past null infinity with the Fock space at future null infinity. So, the nonperturbative late-time corrections to the inner product that Prof. Harlow describes do not appear in flat space. This is the reason we did not need to explicitly discuss Assumption (1).

Turning now to AdS, it is true that the Hilbert space described in (A.8) of our paper is overcomplete due to the corrections that Prof. Harlow notes. However, this does not affect our argument. Our argument only requires that the set of states in A.8 span the Hilbert space. If there are relations between such states, so that two states that appear to be distinct in effective-field theory are identified in the full theory, then these relations would also appear in formula A.17. Of course, we would not be able to see such relations from within low-energy effective field theory but this does not affect the validity of Result 5.

Assumption about the vacuum.

We agree with Prof. Harlow that we could also have assumed that the Hamiltonian of the full theory is a boundary term. Certainly, this is true at all orders in perturbation theory since this property of gravity just follows from diffeomorphism invariance.

Nevertheless, the precise assumption that we need is slightly weaker. We stated this as the assumption that the vacuum could be correctly identified by a boundary term. In fact, it is possible to check that our argument in Lemma 1 would go through even if the state identified by the boundary term was a superposition of finite-energy states in the full theory. It may be the case that even this can only happen if the ADM Hamiltonian is exactly the same as the true Hamiltonian, although the argument of Qi and Ranard does not apply directly to this case. But given how poorly we understand quantum gravity in flat space we would prefer not to state our assumption in a stronger form than we need.

More substantively, it is true that analytic continuation can sometimes turn small terms into large terms. But, we present concrete formulas, both in flat space (Eqn 3.14) and in AdS (A.17) , to build operators in the bulk from near the boundary. At low energies, the operators X_n and X_m can be written down explicitly. At low energies, one can check that if the operator T_{s,s'} in Eqn 3.14 or the operator P_0 in A.17 gets corrected by small nonperturbative terms, this small correction does not become large after being multiplied by the operators X_n and X_m. We agree that it would be interesting to understand whether small corrections in T_{s,s'} or P_0 can become significant if one considers more complicated operators relevant for black holes. This is also relevant if one wants to use Eqn 3.14 and A.17 to set up a physical protocol to identify black-hole microstates since one cannot expect to identify the vacuum exactly.

Page curve

Turning to the Page curve, we would like to make a few separate points.

a) It was sometimes claimed that in flat space the von Neumann entropy of the exact state on I^{+} would follow the Page curve. Our paper points out that this is incorrect: the entropy of a segment of I^{+} is independent of its upper limit. This is not just a mathematical statement; its physical interpretation is that information about the black hole can always be recovered from its exterior when gravity is dynamical.

b) As Prof. Harlow notes, in recent studies, a Page curve has been obtained by coupling the system to a nongravitational bath. The Hilbert space factorizes because gravity switches off in the bath, and so the precise computation that is performed pertains to the transfer of information between two nongravitational systems (one of which may have a holographic dual). It is sometimes tacitly assumed in the literature that this provides a model for the evaporation of more realistic black holes, including black holes in asymptotically flat space. Our work raises serious questions about the validity of such an extrapolation. We note that this is supported by the results of arXiv:2012.04671, where it was shown that if one turns on gravity in the bath, the Page curve disappears.

c) Page's original suggestion that the entropy of radiation should rise and then fall was based on the incorrect assumption that one should be able to effectively factorize the Hilbert space even with dynamical gravity. This is such an appealing assumption that it has been made repeatedly in the literature. But, it is necessary to be very careful with such an assumption since it has often led to misleading results and paradoxes.

Prof. Harlow suggests that one can obtain a useful notion of factorization by considering approximate operators in the little Hilbert space or code subspace. But we do not understand his claim that bulk locality can be made much more accurate in the background of a small AdS black hole than e^{-S}. Perhaps Prof. Harlow is referring to operators confined to separate entanglement wedges. An operator in one entanglement wedge can be made to commute to high accuracy with an operator in another entanglement wedge by dressing the two operators to separate sections of the boundary. But the geometry relevant for black hole evaporation is different. The black hole is surrounded by its radiation, and so the interior and the radiation do not live in separate entanglement wedges. There is no way to dress an operator in the black hole interior to the boundary, without having the gravitational Wilson line run through the radiation. So in this situation we do not see how to make the commutator of operators in the interior and the exterior any smaller than e^{-S}.

This is the reason that we argue that the failure of factorization is not a technicality but affects the physics.

d) Prof. Harlow's asks about what a quantum computer that collects the radiation would tell us. In the paper we state that such questions are "imprecise" because one can get different answers by endowing the quantum computer with different abilities.

For instance if the quantum computer is sensitive only to low-point correlators then we expect see the Hawking curve (which rises monotonically). If the quantum computer is sensitive to all physical degrees of freedom, including metric fluctuations and exponentially small effects, then we expect to see the flat Page curve.

Note that we do not rule out the possibility that at some intermediate level of coarse-graining, one may see the conventional Page curve. In the section on "the Page curve in flat space" we even discuss some concrete possibilities for this. This is because once one has all information in hand by measuring all operators, it should be possible to "discard information" at just the right rate to see the Page curve.

In summary, we do not believe that the Page curve describes the fine-grained entropy of black-hole radiation in situations where gravity is everywhere dynamical. But even when gravity is dynamical, we agree it may be possible to ask a different question so that the answer is the Page curve, although it remains to be seen whether such a question would be natural from a physical point of view.

---

## Round 2 · Referee Report · Anonymous (Referee 2) · 2021-1-30

Strengths

The paper is well written and, in my opinion, it makes an important contribution in making it precise some computations, arguments and hypothesis that are usually vaguely discussed and/or implicitly assumed in the literature. This contribution can be thought of as a thoughtful discussion about certain features of non quantum gravity that are related to holography.

Weaknesses

While I consider that the paper contains some claims that could be taken as over-claiming by some colleagues working in the subject, I think it does contribute to the on-going discussion on the Page curve, the holographic realization of black hole evaporation, and so I consider it worth publishing.

Report

The paper is well written and, in my opinion, it makes an important contribution in making it precise some computations, arguments and hypothesis that are usually vaguely discussed and/or implicitly assumed in the literature. This contribution can be thought of as a thoughtful discussion about certain features of non quantum gravity that are related to holography. While I consider that the paper contains some claims that could be taken as over-claiming by some colleagues working in the subject, I think it does contribute to the on-going discussion on the Page curve, the holographic realization of black hole evaporation, and so I consider it worth publishing.

Requested changes

I find the paper worthwhile publishing as it is.

  • validity: high
  • significance: high
  • originality: high
  • clarity: top
  • formatting: excellent
  • grammar: excellent

Author:  Suvrat Raju  on 2021-02-05  [id 1205]

(in reply to Report 2 on 2021-01-30)

We would like to thank the referee for the summary of the paper and for the referee's kind comments.

For the benefit of readers, we would only like to reiterate that, as discussed in section 4.2, there is no contradiction between our results and the recent literature on the Page curve. The Page curve has been obtained by coupling a holographic system to a nongravitational bath. Our results only imply that this calculation is not directly relevant for more realistic models of black-hole evaporation, where gravity is dynamical everywhere, or for black-hole evaporation in asymptotically flat space.

This issue is also discussed in more detail in our reply to report 1.

We would also like to refer readers to arXiv:2012.04671 where it is argued, from a different perspective, that the Page curve does not describe the entropy of the radiation when gravity is dynamical everywhere, although it may still be the answer to other questions. The absence of the Page curve is related to the physical statement that information about the interior can be obtained outside the black hole when gravity is dynamical. In this context, we would also like to refer readers to arXiv:2008.01740 where the idea that information about a region can be extracted by distant observers using gravitational effects is concretely verified in the low-energy theory.

---

## Round 2 · Referee Report · Anonymous (Referee 3) · 2021-2-1

Strengths

Clear presentation, self-contained, interesting implications.

Weaknesses

Somewhat strong claims at the end of section 4.

Report

This paper raises some very interesting questions about the holographic nature of asymptotically flat spacetimes. The presentation is clear and the result was somewhat surprising so I anticipate their will be a rich interplay between what the authors have shown and how we interpret celestial CFT correlators.

I think the concerns regarding the other Referee's "assumption 1)" are safer in the context of fields at null infinity. I am less confident about how the authors would extend this to incorporate massive fields. So too, some of the claims about implications for black hole evaporation seem a tad bit strong and not clearly decoupled from their starting assumptions.

Requested changes

N/A

  • validity: good
  • significance: high
  • originality: good
  • clarity: high
  • formatting: perfect
  • grammar: perfect

Author:  Suvrat Raju  on 2021-02-05  [id 1206]

(in reply to Report 3 on 2021-02-01)

We would like to thank the referee for their report.

We agree with the referee that the construction of the Hilbert space can be performed cleanly at null infinity, and we have also noted the referee's point in our response to report 1. We also agree that it would be very interesting to extend our results to massive particles, and we hope to be able to report concrete progress on this front in the near future.

For further discussion of section 4, we refer readers to our reply to Reports 1 and 2.

---

## Editorial Decision

published